# Combining DNA and HPTLC profiles to differentiate a pain relief herb, *Mallotus repandus*, from plants sharing the same common name, "Kho-Khlan"

**Kannika Thongkhao¤, Chayapol Tungphatthong, Vipawee Pichetkun, Suthathip Gaewtongliam, Worakorn Wiwatcharakornkul, Suchada Sukrong*

Faculty of Pharmaceutical Sciences Chulalongkorn University, Department of Pharmacognosy and Pharmaceutical Botany, Center of Excellence in DNA Barcoding of Thai Medicinal Plants, Bangkok, Thailand

¤ Current address: School of Languages and General Education, Walailak University, Nakhon Si Thammarat, Thailand
* suchada.su@chula.ac.th

**Data Availability Statement:** All relevant data are within the paper and its Supporting Information files.

## Abstract

The pain relief formula "Ya Pa Som Kho-Khlan (YPSKK)" or "ยาผสมโคคลาน°" in Thai is officially recorded in the Natural List of Essential Medicines (NLEM) of Thailand. The main component is *Mallotus repandus* (Willd.) Müll. Arg.; however, *Anamirta cocculus* (L.) Wight & Arn and *Croton caudatus* Gleiseler share the same common name: "Kho-Khlan". Confused usage of *A. cocculus* or *C. caudatus* can have effects via toxicity or unsuccessful treatment. This study aimed to combine a high-performance thin-layer chromatography (HPTLC) technique and DNA barcoding coupled with high-resolution melting (Bar-HRM) to differentiate *M. repandus* from the other two species. The *M. repandus* extract exhibited a distinct HPTLC profile that could be used to differentiate it from the others. DNA barcodes of the *rbc*L, *mat*K, ITS and *psb*A-*trn*H intergenic spacer regions of all the plants were established to assist HPTLC analysis. The *rbc*L region was selected for Bar-HRM analysis. PCR amplification was performed to obtain 102 bp amplicons encompassing nine polymorphic nucleotides. The amplicons were subjected to HRM analysis to obtain melting curve profiles. The melting temperatures ($T_m$) of authentic *A. cocculus* (A), *C. caudatus* (C) and *M. repandus* (M) were separated at 82.03±0.09˚C, 80.93±0.04˚C and 80.05±0.07˚C, respectively. The protocol was applied to test crude drugs (CD1-6). The HPTLC profiles of CD2-6 showed distinct bands of *M. repandus*, while CD1 showed unclear band results. The Bar-HRM method was applied to assist the HPTLC and indicated that CD1 was *C. caudatus*. While ambiguous melting curves from the laboratory-made formulae were obtained, HPTLC analysis helped reveal distinct patterns for the identification of the plant species. The combination of HPTLC and Bar-HRM analysis could be a tool for confirming the identities of plant species sharing the same name, especially for those whose sources are multiple and difficult to identify by either chemical or DNA techniques.

**Funding:** The authors received no specific funding for this work.

**Competing interests:** The authors have declared that no competing interests exist.

## Introduction

Common name sharing among herbal species can cause confusion during herbal medicine preparation, leading to less efficient treatment and undesirable effects due to improper therapeutic potential. In Thailand, the traditional herbal formula used for pain relief is called "ยา ผสมโคคลาน" in Thai or "Ya Pa Som Kho-Khlan (YPSKK)", which is officially recorded in the National List of Essential Medicines (NLEM), an official national standard compendium. According to the NLEM, YPSKK is a mixed herbal formula consisting of *Mallotus repandus* (Willd.) Müll. and three other species, *Elephantopus scaber* L., *Aegle marmelos* (L.) Corrêa and *Rhinacanthus nasutus* (L.) Kurz [1–3]. *M. repandus* (Euphorbiaceae) shares the common name "Kho-Khlan" with *Croton caudatus* Gleiseler (Euphorbiaceae) and *Anamirta cocculus* (L.) Wight & Arn (Menispermaceae) (Fig 1). However, only *M. repandus* (Fig 1A) is an official plant species in NLEM.

The stem of *M. repandus* has long been used for the relief of muscle pain in Thai traditional medicine [2]. *C. caudatus* is administered for headaches, visceral pain, rheumatism, fever, and constipation [4–6]. The crude extract of *C. caudatus* seeds can protect against mosquito larvae [7]. *A. cocculus* is used in the treatment of blood stasis and fever, stimulates the central nervous system [8] and is recorded as a restorative medical herb in the southern region of Thailand [9]. However, a previous report showed that *C. caudatus* causes irritation and allergic responses [10], while *A. cocculus* contains very strong neurotoxin compounds that affect the central nervous system (CNS) of vertebrates, such as picrotoxin, picrotin, methyl picrotoxate, dihydroxy-picrotoxinin, picrotoxic acid and a sesquiterpene mixture of picrotoxinin [11–13]. Seeds of *A. cocculus* are also used to eliminate unwanted wild fish in aquaculture ponds and to kill birds [14]. Consuming *A. cocculus* berries causes extensive brain hemorrhage in cattle, while small amounts of *A. cocculus* are highly toxic and fatal if consumed by humans [11,13]. Although the substances in *A. cocculus* are harmful, the herb is still utilized in Thai traditional medicine due to the belief that a very small dose of toxic substances can be neutralized by other compounds in the herbal formula [15].

The stem of *M. repandus* is used for the YPSKK formula. Crude drugs of *M. repandus* are commercially provided in both powdered form and small pieces of stem, which are challenging for species differentiation (Fig 1B and 1C). Although raw materials of herbal medicine can be examined by simple organoleptic methods and macroscopic and microscopic methods, experienced personnel for taxonomic examination are required [16]. Thin-layer chromatography (TLC) and high-performance TLC (HPTLC), which are recommended in the herbal pharmacopoeias of many countries, including Thailand, are reliable methods for phytochemical constituent examination; however, the methods require a target compound as a standard reference [17,18]. HPTLC, a sophisticated form of TLC, provides good separation efficiency due to the higher quality of its separation plate. HPTLC exhibits higher accuracy, reproducibility, and ability to document the results than TLC [18]. Therefore, this method has been used to determine the phytochemical profile of herbal species. However, uncertain results may occur due to environmental factors that affect the chemical composition of herbal species and biological activities of the substances [19]. In recent years, a molecular approach called the DNA barcoding technique has gained demand in species identification because it is an accurate, cost-effective and reliable tool for species identification. The DNA barcoding method provides species-level information, and small amounts of samples are needed for the identification process [19].

Currently, DNA barcoding coupled with high-resolution melting (Bar-HRM) analysis has gained attention for its rapid identification of herbal species such as *Vaccinium myrtillus* L. [20], *Mitragyna speciosa* Korth [21] and *Ardisia gigantifolia* Stapf [22]. Bar-HRM, a

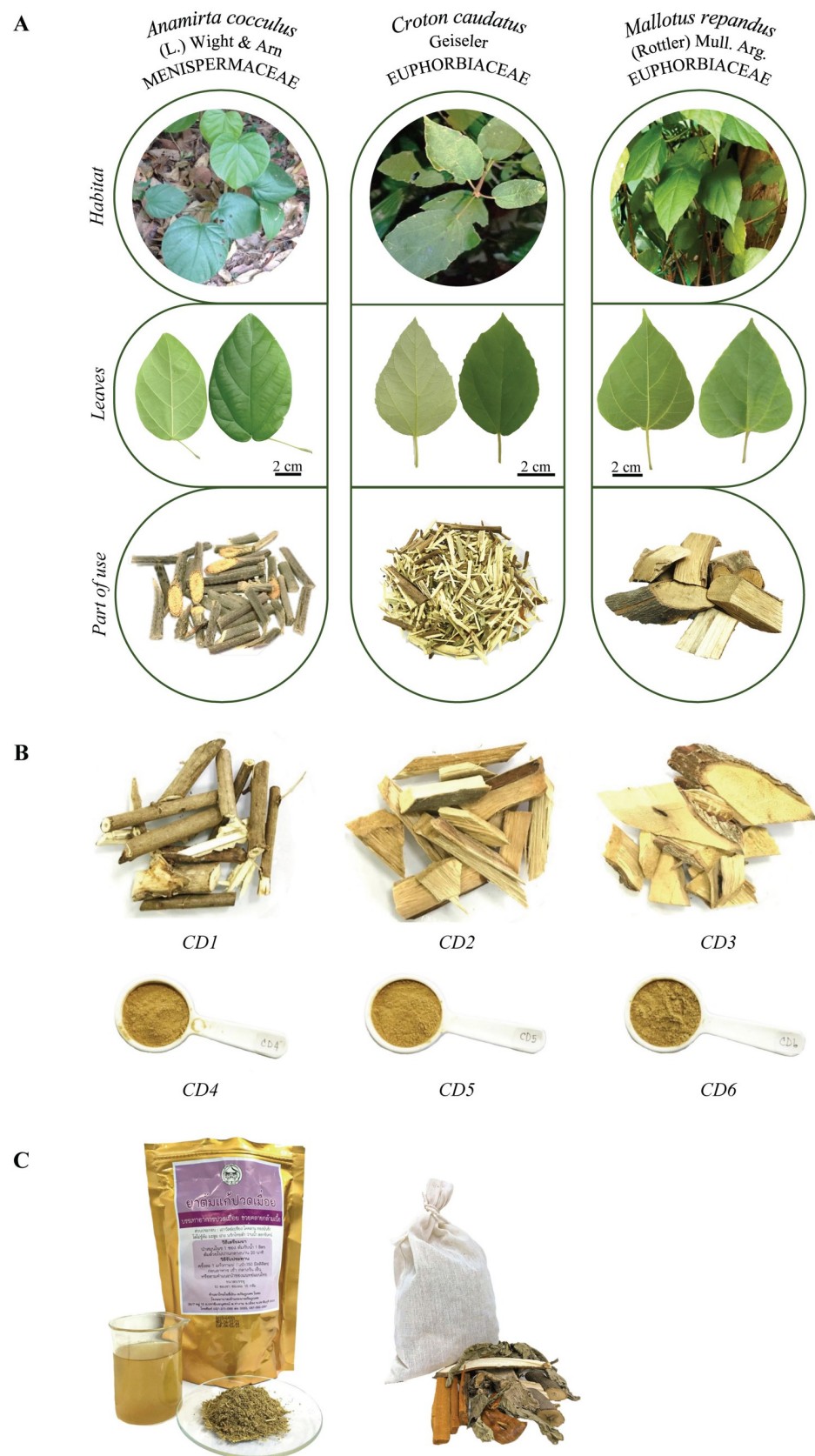

**Fig 1. Samples used in this study.** (A) Authentic plant species: *Anamirta cocculus* (L.) Wight & Arn, *Croton caudatus* Geiseler and *Mallotus repandus* (Rottler) Müll. Arg., (B) purchased crude drug samples called Kho-Khlan: CD1-CD6, and (C) commercial YPSKK formula.

sequencing-free method, detects signal alterations during the dissociation of double-stranded DNA generated from the PCR into single-stranded DNA. Each plant species can be differentiated by their individual melting temperature ($T_m$), which is correlated to their nucleotide sequences in the target region [23]. Bar-HRM analysis is a fast, cost-effective and reliable method; moreover, a small amount of sample is required for species identification. However, Bar-HRM primer design is challenging when the target sequence has high variation rates across the target amplicon, and Bar-HRM analysis is limited when low-quality DNA templates are used [24].

As mentioned above, each identification method has advantages and limitations; therefore, an integrative approach is proposed to differentiate substitutions or adulterants of herbal species [19,25]. Combined phytochemical profiles and DNA information can be applied to prevent the use of incorrect herbal species and support the quality of herbal materials to meet international standards [19]. In this study, we aimed to utilize HPTLC and Bar-HRM analysis to differentiate a pain relief herb, *M. repandus*, from *C. caudatus* and *A. cocculus*, which share the common name Kho-Khlan. Combined approaches were used to create a simple and rapid identification method for the quality control of the Kho-Khlan raw material in the herbal industry.

## Materials and methods

### Plant materials

Fresh leaves and stems of *A. cocculus* (n = 8), *C. caudatus* (n = 8) and *M. repandus* (n = 8) were collected from various locations across Thailand (Table 1). These collections are legally permitted. The plant samples were identified by a taxonomist, Associate Professor Chaiyo Chaichantipyuth, at the Department of Pharmacognosy and Pharmaceutical Botany of Chulalongkorn University. All voucher specimens were deposited at the Center of Excellence in DNA Barcoding of Thai Medicinal Plants, Chulalongkorn University, Thailand. Six commercial crude drugs claiming to be Kho-Khlan were purchased from local stores in Thailand. The three plant ingredients, *E. scaber*, *A. marmelos* and *R. nasutus*, in YPSKK were purchased from local dispensaries. All experiments were performed in accordance with relevant guidelines and regulations.

### Preparation of herbal mixture samples and laboratory-made YPSKK formulae

Mixtures of (i) *A. cocculus* and *C. caudatus*, (ii) *A. cocculus* and *M. repandus*, and (iii) *C. cocculus* and *M. repandus* were prepared. Briefly, 100 g of *A. cocculus*, *C. caudatus* and *M. repandus* stems were weighed and ground into fine powders. The powder from each species was mixed in different proportions as follows: 10:90, 25:75 and 50:50 (w/w). A three-species mixture of *A. cocculus*, *C. caudatus*, and *M. repandus* (iv) was also made at a mixing ratio of 1:1:1.

Laboratory-made YPSKK formulae were created according to the plant species listed in the NLEM of Thailand. The ingredient-based powder was prepared by mixing equal amounts of *E. scaber*, *A. marmelos* and *R. nasutus* (mixing ratio 1:1:1). Then, 3 g of base powder was combined with 1 g of *A. cocculus* (A), *C. caudatus* (C) and *M. repandus* (M) powders to create an *A. cocculus*-containing formula (F-A), a *C. caudatus*-containing formula (F-C) and a *M.*

**Table 1. Samples used in this study along with their DNA barcode locus accession numbers in GenBank.** Crude drugs claiming to be Kho-Khlan purchased from local markets and laboratory-made formulae are listed.

| Plant species | Voucher number | Collection location | Accession number | | | |
|---|---|---|---|---|---|---|
| | | | ITS | *mat*K | *rbc*L | *psb*A-*trn*H |
| **Authentic species** | | | | | | |
| *Anamirta cocculus* (L.) Wight & Arn | SS-579 | Bangkok | LC506294 | LC506295 | LC506296 | LC506297 |
| | SS-583 | Nakhonnayok | LC506306 | LC506307 | LC506308 | LC506309 |
| | SS-587 | Chanthaburi | LC506318 | LC506319 | LC506320 | LC506321 |
| | SS-622 | Chanthaburi | LC506330 | LC506331 | LC506332 | LC506333 |
| | SS-706 | Nakhonnayok | LC506342 | LC506343 | LC506344 | LC506345 |
| | SS-707 | Nakhonnayok | LC506354 | LC506355 | LC506356 | LC506357 |
| | SS-711 | Bangkok | LC506366 | LC506367 | LC506368 | LC506369 |
| | SS-712 | Bangkok | LC506378 | LC506379 | LC506380 | LC506381 |
| *Croton caudatus* Gleiseler | SS-537 | Bangkok | LC506286 | LC506287 | LC506288 | LC506289 |
| | SS-588 | Nonthaburi | LC506298 | LC506299 | LC506300 | LC506301 |
| | SS-589 | Ubonratchathani | LC506310 | LC506311 | LC506312 | LC506313 |
| | SS-628 | Bangkok | LC506322 | LC506323 | LC506324 | LC506325 |
| | SS-715 | Nonthaburi | LC506334 | LC506335 | LC506336 | LC506337 |
| | SS-716 | Prachinburi | LC506346 | LC506347 | LC506348 | LC506349 |
| | SS-717 | Prachinburi | LC506358 | LC506359 | LC506360 | LC506361 |
| | SS-718 | Bangkok | LC506370 | LC506371 | LC506372 | LC506373 |
| *Mallotus repandus* Müll. Arg. | SS-538 | Bangkok | LC506290 | LC506291 | LC506292 | LC506293 |
| | SS-590 | Ratchaburi | LC506302 | LC506303 | LC506304 | LC506305 |
| | SS-667 | Bangkok | LC506314 | LC506315 | LC506316 | LC506317 |
| | SS-708 | Yasothon | LC506326 | LC506327 | LC506328 | LC506329 |
| | SS-709 | Yasothon | LC506338 | LC506339 | LC506340 | LC506341 |
| | SS-710 | Prachinburi | LC506350 | LC506351 | LC506352 | LC506353 |
| | SS-713 | Ubonratchathani | LC506362 | LC506363 | LC506364 | LC506365 |
| | SS-714 | Nakhonnayok | LC506374 | LC506375 | LC506376 | LC506377 |
| **Crude drugs** | | | | | | |
| Crude drug 1 (CD1) | SS-777 | Bangkok | - | - | - | - |
| Crude drug 2 (CD2) | SS-778 | Ubonratchathani | - | - | - | - |
| Crude drug 3 (CD3) | SS-779 | Bangkok | - | - | - | - |
| Crude drug 4 (CD4) | SS-780 | Bangkok | - | - | - | - |
| Crude drug 5 (CD5) | SS-781 | Yasothon | - | - | - | - |
| Crude drug 6 (CD6) | SS-782 | Nakhonnayok | | | | |
| **Herbal formulae** | | | | | | |
| *A. cocculus*-containing formula (F-A) | SS-783 | Bangkok | - | - | - | - |
| *C. caudatus*-containing formula (F-C) | SS-784 | Bangkok | - | - | - | - |
| *M. repandus*-containing formula (F-M) | SS-785 | Bangkok | - | - | - | - |
| Mixed formula of *A. cocculus*, *C. caudatus* and *M. repandus* (F-ACM) | SS-786 | Bangkok | - | - | - | - |

*repandus*-containing formula (F-M), respectively. One gram of mixed Kho-Khlan plants, including *A. cocculus*, *C. caudatus* and *M. repandus*, was combined with 3 g of base powder to generate a three-plant mixture (F-ACM).

## HPTLC profiles

To obtain the phytochemical profiles of selected samples, including *A. cocculus* (SS-628), *C. caudatus* (SS-537) and *M. repandus* (SS-583), 1 g of dried stems from each species was crushed

into a fine powder using a M 20 Universal mill grinder (IKA, Germany). Phytochemical constituents were extracted in ethanol (1:20, w/v) at room temperature. The solution was mixed with a vortex mixer for 30 s and subsequently incubated in an ultrasonic bath for 15 min at room temperature. The supernatant was collected after centrifugation at 10,000 rpm for 10 min at 25°C. Then, 5 μl of the extracted solution was spotted onto an HPTLC plate (20×10 cm, Silica gel 60 $F_{254}$, Merck, Germany) using an Automatic TLC Sampler 4 (AST4, CAMAG, Muttenz, Switzerland). Each individual band was 8 mm in length, the distance between tracks was 2 mm, and the track distance was 11.4 mm from the lower edge of the plate. The distance from the left side was 16 mm, and the distance from the lower edge was 20 mm. Toluene:acetone:formic acid (5:4:0.5, v/v/v) was used as the mobile phase. The chamber was saturated with 20 ml of mobile phase for 20 min before development. HPTLC plate visualization was performed under ultraviolet light at short and long wavelengths of 254 nm and 366 nm, respectively. The HPTLC method was applied to test commercial Kho-Khlan crude drugs and the laboratory-made YPSKK formulae. The extraction protocol and HPTLC method were as previously mentioned.

## Genomic DNA extraction

Genomic DNA from leaves of the samples, the purchased crude drugs, mixed herbal powder and laboratory-made YPSKK formulae were extracted using a DNeasy Plant Mini Kit (Qiagen, Germany) and further purified using a GENECLEAN Kit (MP Biomedicals, USA) according to the manufacturer's protocol. DNA quantity and quality were determined using a NanoDrop One UV–Vis Spectrophotometer (Thermo Scientific, USA) and agarose gel electrophoresis, respectively. Genomic DNA was run on 0.8% (w/v) agarose in 1X TAE gel containing 1X Red-Safe nucleic acid staining solution (iNtRON Biotechnology, USA) at 100 V for 30 min. Agarose gel was analyzed with a UVP GelSolo (Analytik Jena GmbH, Germany) gel documentation system, and images were taken by onboard VisionWorks software (Analytik Jena GmbH, Germany). Genomic DNA was stored at -20°C for further use.

## DNA barcoding of *A. cocculus*, *C. caudatus* and *M. repandus*

Genomic DNA from the leaves of *A. cocculus*, *C. caudatus* and *M. repandus* was used as a DNA template for DNA barcode generation. The following DNA barcode regions were amplified by the primers listed in Table 2: maturase K (*mat*K), the large subunit of ribulose-1,5-bisphosphate carboxylase/oxygenase (*rbc*L), the *trn*H-*psb*A intergenic spacer and the nuclear internal transcribed spacer (ITS). PCR amplification was performed in a 50 μl reaction mixture. The PCR mixture contained 1X PCR buffer with 1.5 mM $MgCl_2$, 0.2 mM dNTP mix, 0.5 μM each forward and reverse primer and 0.5 U of Platinum *Taq* DNA polymerase (Invitrogen, USA). Fifty nanograms of genomic DNA was used as the DNA template. PCR was carried out in a GS-96 Gradient Touch Thermal Cycler (Hercuvan, UK) using cycling conditions of 94°C for 4 min followed by 30 cycles of 94°C for 30 sec, 57°C for 30 sec, and 72°C for 1:30 min (*rbc*L and *mat*K) or 45 sec (for ITS and *psb*A-*trn*H spacer) and a final extension at 72°C for 10 min. The amplified products were determined on a 1.2% (w/v) agarose gel in 1X TAE buffer containing 1X RedSafe nucleic acid staining solution. Agarose gel analysis was performed as described above. PCR products were further sequenced by direct sequencing in both directions on an ABI 3730XL DNA analyzer using the primers listed in Table 2 and S1 Appendix. The sequencing results were analyzed by Molecular Evolutionary Genetics Analysis X (MEGA X) software version 10.1. The DNA barcode sequences were deposited in GenBank of the National Center for Biotechnology Information (NCBI) (Table 1).

**Table 2. Primers for DNA barcode generation and Bar-HRM analysis.**

| Barcode region | Primer name | Primer sequence (5′-3′) | References |
|---|---|---|---|
| **DNA barcode generation** | | | |
| *rbc*L | *rbc*L_aF | ATGTCACCACAAACAGAGACTAAAGC | Levin et al., 2003 |
| | *rbc*L-R23 | TTTTAGTAAAAGATTGGGCCG | Ohi-Toma et al., 2006 |
| *mat*K | *trn*K-3914F | TGGGTTGCTAACTCAATGG | Johnson et al., 1994 |
| | *trn*K-2R | AACTAGTCGGATGGAGTAG | Johnson et al., 1994 |
| | *mat*K-aF | CTATATCCACTTATCTTTCAGGAGT | Kato et al., 1999 |
| | *mat*K-8R | AAAGTTCTAGCACAAGAAAGTCGA | Kato et al., 1999 |
| *trn*H-*psb*A | *psb*A_*trn*HF | GTTATGCATGAACGTAATGCTC | Sang et al., 1997 |
| | *psb*A-*trn*HR | CGCGCATGGTGGATTCACAATC | Sang et al., 1997 |
| ITS | ITS1 | TCCGTAGGTGAACCTGCGG | White et al., 1990 |
| | ITS4 | TCCTCCGCTTATTGATATGC | White et al., 1990 |
| **Bar-HRM primers** | | | |
| *rbc*L | KK-rbcL-HRM-F | TTTCACTCAAGATTGGGTCTCT | This study |
| | KK-rbcL-HRM-R | TCATCTCCAAAGATCTCGGTCA | This study |

## Differentiation of *M. repandus* from *A. cocculus* and *C. caudatus* by Bar-HRM analysis

To design Bar-HRM primers, nucleotide sequences obtained from the four DNA barcode regions of *M. repandus*, *A. cocculus* and *C. caudatus* were aligned by MUSCLE with gap open = −400; gap extend = 0; clustering method = UPGMB; and Min Diag Length = 24. The *rbc*L region was selected to perform Bar-HRM analysis for the differentiation of *M. repandus* from *A. cocculus* and *C. caudatus*. Primer 3 and BLAST software were used for primer design. The Bar-HRM forward (KK-rbcL-HRM-F) and reverse primers (KK-rbcL-HRM-R) were designed based on the conserved regions of the *rbc*L gene of the three plants. The targeted amplicon provided a 102 bp amplicon with 9 polymorphic sites of the *rbc*L gene. PCR amplification was performed in a total volume of 10 µl on a CFX96 Real-time System (Bio–Rad, USA). The reaction mixture contained 10 ng of genomic DNA, 1X SsoFast EvaGreen Supermix (Bio–Rad, USA), 0.5 µM forward primer (KK-rbcL-HRM-F: 5′-TTTCACTCAAGATTGGGTCTCT—3′) and reverse primer (KK-rbcL-HRM-R: 5′-TCATCTCCAAAGATCTCGGTCA-3′). Real-time PCR conditions were as follows: initial denaturing step at 95˚C for 1 min followed by 39 cycles of 95˚C for 15 sec, 60˚C for 15 sec, and 72˚C for 15 sec. Subsequently, the PCR amplicons were denatured at 9˚C for 1 min and reannealed at 60˚C for 1 min to generate random DNA duplexes. Melting curves ($T_m$) were generated after the last extension step. The temperature was set to increase from 60˚C to 95˚C in 0.1˚C increments, and the fluorescence intensity was collected at each increasing step. CFX Manager software (version 3.1 upgrade) and Precision Melt Analysis software (version 3.1 upgrade) were used to analyze the $T_m$. Normalized curves and differential melting curves were plotted. *C. caudatus* was set as the reference species. Reactions were performed in triplicate. Sensitivity was analyzed using genomic DNA at different concentrations: $10 \times 10^{-9}$, $1 \times 10^{-9}$, $0.1 \times 10^{-9}$, $0.01 \times 10^{-9}$ and $0.001 \times 10^{-9}$ g.

## Bar-HRM analysis of purchased crude drugs, plant mixtures and laboratory-made YPSKK formulae

The Bar-HRM method was applied to test the authenticity of the six commercial crude drugs claiming to be Kho-Khlan. The method was conducted to identify herbal species within plant mixtures and four laboratory-made YPSKK formulae. The Bar-HRM reaction and conditions

were as mentioned above. The Bar-HRM analysis parameters were set as described earlier. All reactions were performed in triplicate.

## Results

### Species-specific patterns of HPTLC

HPTLC profiles from ethanolic extracts of *A. cocculus*, *C. caudatus* and *M. repandus* were obtained. Species-specific bands were obtained from authentic *A. cocculus* (Rf = 0.22), *C. caudatus* (Rf = 0.02 and 0.60) and *M. repandus* species (Rf = 0.08, 0.26, 0.68 and 0.72). The HPTLC profiles of six crude drugs claiming to be Kho-Khlan were compared to those of authentic plants. In general, the HPTLC patterns among CD2-CD6 were similar. A bright blue band at Rf = 0.72 for *M. repandus* was found in CD2-CD6. The blue band at Rf = 0.08 was present in CD2-CD6, while the band at Rf = 0.26 was present in CD2-CD6. The crude drug CD1 showed an ambiguous HPTLC pattern with a faint band at Rf = 0.02. No distinct bands of *A. cocculus* (Rf = 0.22) or *C. caudatus* (Rf = 0.60) were detected in any of the crude drug samples (Fig 2).

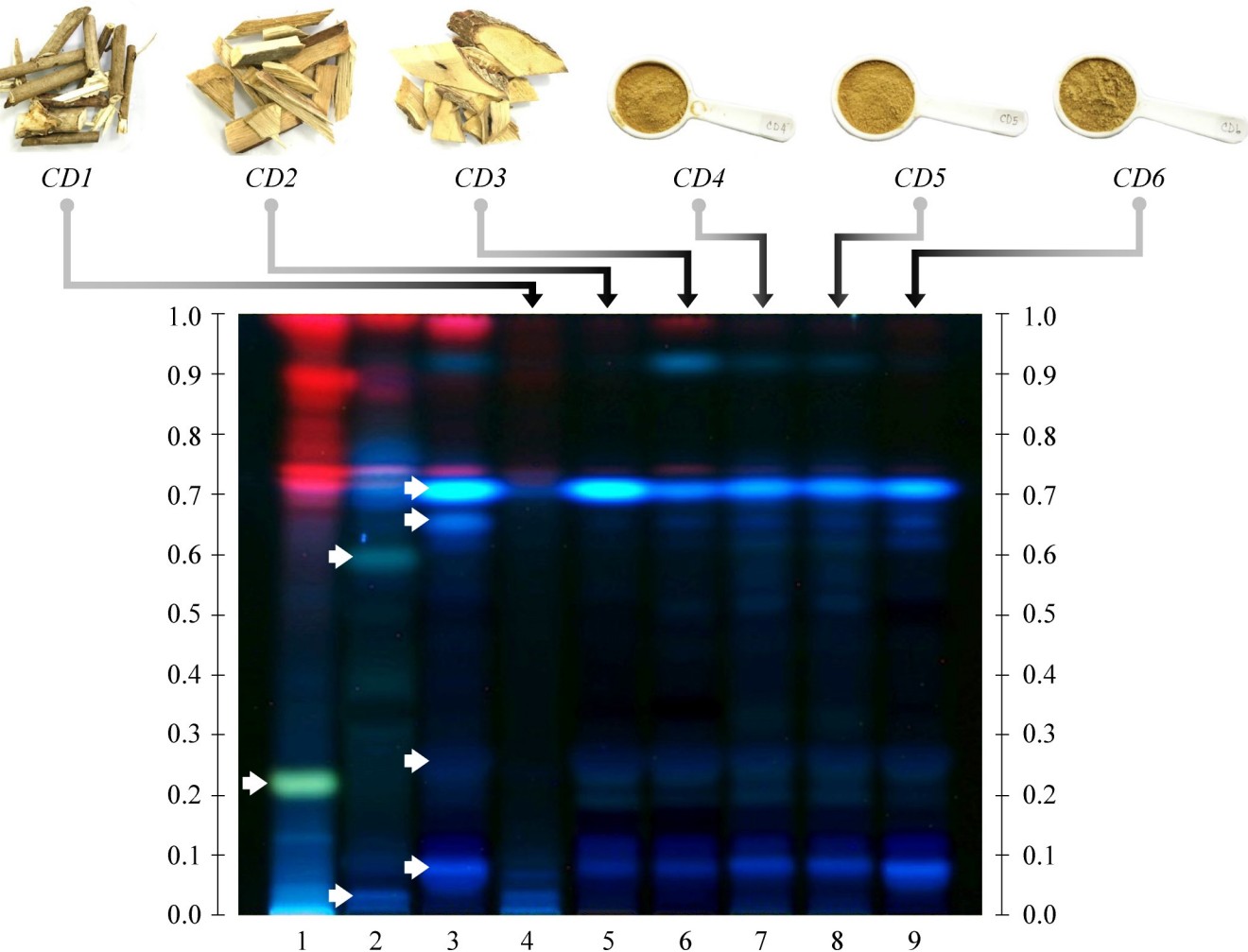

**Fig 2. High-performance thin-layer chromatography (HPTLC) chromatogram of ethanolic extracts under UV at 366 nm.** Track 1: *A. cocculus*, track 2: *C. caudatus*, track 3: *M. repandus*, tracks 4–9: Crude drugs CD1-CD6. A toluene:acetone:formic acid mixture (5:4:0.5, v/v/v) was used as the mobile phase. White arrows indicate the characteristic bands of each authentic plant species.

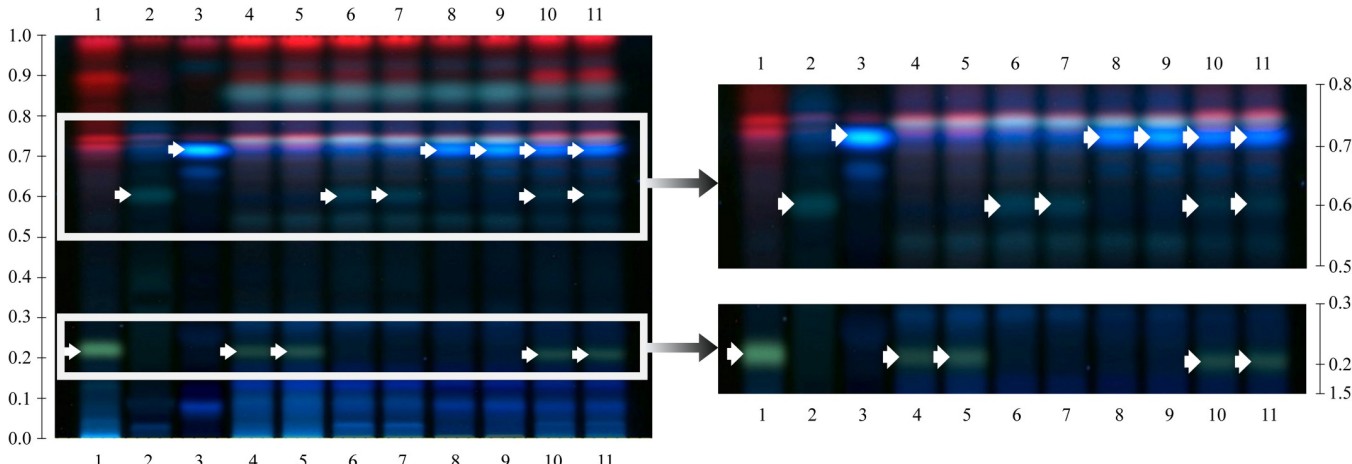

**Fig 3. HPTLC chromatograms of ethanolic extracts of authentic species and four laboratory-made YPSKK formulae under UV at 366 nm.** Track 1: *A. cocculus*, track 2: *C. caudatus*, track 3: *M. repandus*, tracks 4–5: F-A, tracks 6–7: F-C, tracks 8–9: F-M and tracks 10–11: F-ACM. A toluene:acetone:formic acid (5:4:0.5, v/v/v) mixture was used as the mobile phase. White arrows indicate the characteristic bands of each plant species.

## HPTLC profiles of plant species in the YPSKK formulae

HPTLC bands unique to *A. cocculus* (Rf = 0.22), *C. caudatus* (Rf = 0.60) and *M. repandus* (Rf = 0.72) were found in F-A, F-C and F-M, respectively. In the F-ACM formula, species-specific bands of *A. cocculus* (Rf = 0.22), *C. caudatus* (Rf = 0.02 and 0.60) and *M. repandus* (Rf = 0.08, 0.68 and 0.72) were detected. Other species-specific bands of *M. repandus* (Rf = 0.26) did not appear in the laboratory-made formulae (Fig 3).

## Establishment of the four core DNA barcode regions

Core DNA barcode regions, including *mat*K, *rbc*L, the *psb*A-*trn*H intergenic spacer and the ITS of *A. cocculus*, *C. caudatus* and *M. repandus*, were successfully amplified and sequenced. Full-length nucleotide sequences were obtained and submitted to GenBank (Table 1). Plant species collected from different locations exhibited identical nucleotide sequences in each DNA barcode region. The lengths of the *rbc*L, *mat*K, ITS and *psb*A-*trn*H intergenic spacer regions were 1428, 1521–1536, 548–635 and 445–783 bp, respectively. Sequence length, GC content (%) and the percentage of variable nucleotide sites varied among the three species. In terms of nucleotide variation, the four DNA barcodes were ranked as follows: ITS (48.71%) > *psb*A-*trn*H intergenic spacer (38.27%) > *rbc*L (37.17%) and *mat*K (23.15%) (Table 3). Nucleotide alignment results for the three species revealed insertions-deletions (indels) within the *mat*K, ITS and *psb*A-*trn*H intergenic spacer regions (S2 Appendix).

## Differentiation of *M. repandus* from *A. cocculus* and *C. caudatus* by Bar-HRM analysis

To conduct the PCR-Bar-HRM analysis, PCR amplification of the *rbc*L gene using Bar-HRM primers encompassing nine nucleotide polymorphic sites was performed in *M. repandus*, *A. cocculus* and *C. caudatus*. A 102 bp PCR amplicon (positions 1,089–1,191) was obtained from all three plant species (Fig 4). HRM analysis was performed to determine the melting temperatures ($T_m$) of each amplicon generated from Bar-HRM primers (Fig 5). Three distinct categories of melting curve profiles, -d(RFU)/dT (Fig 5A), normalized RFU (Fig 5B) and different RFU (Fig 5C), were clearly detected among the three plants. The $T_m$ values of *A. cocculus*, *C. caudatus* and *M. repandus* were 82.03 ± 0.09°C, 80.93 ± 0.04°C and 80.05 ± 0.07°C, respectively

**Table 3. Sequence analysis of core DNA barcode regions of *A. cocculus*, *C. caudatus* and *M. repandus*.**

| Region | Species | Properties | | |
|---|---|---|---|---|
| | | Length (bp) | GC content (%) | Variability (%) |
| ITS | *A. cocculus* | 548 | 56.20 | 48.71 |
| | *C. caudatus* | 626 | 56.23 | |
| | *M. repandus* | 635 | 60.00 | |
| *mat*K | *A. cocculus* | 1536 | 33.30 | 23.15 |
| | *C. caudatus* | 1521 | 30.97 | |
| | *M. repandus* | 1521 | 30.37 | |
| *rbc*L | *A. cocculus* | 1428 | 44.68 | 37.17 |
| | *C. caudatus* | 1428 | 43.63 | |
| | *M. repandus* | 1428 | 43.42 | |
| *psb*A-*trn*H | *A. cocculus* | 640 | 29.53 | 38.27 |
| | *C. caudatus* | 445 | 25.62 | |
| | *M. repandus* | 783 | 21.97 | |

(Table 4). Gold-standard Sanger sequencing of PCR amplicons and blast analysis confirmed the original species of each amplicon. Similar melting profiles among different DNA concentrations ($10 \times 10^{-9}$, $1 \times 10^{-9}$, $0.1 \times 10^{-9}$ and $0.01 \times 10^{-9}$ g) were revealed for *M. repandus*. The melting curves displayed small changes and isolated clusters at concentrations of $0.001 \times 10^{-9}$ and $0.0001 \times 10^{-9}$ g when analyzed using Precision Melt Analysis software (S3 Appendix).

## Application of Bar-HRM for the identification of herbal materials

Six crude drugs (CD1-CD6) were investigated to identify their botanical species. CD1 exhibited a melting temperature of 80.84±0.06°C, which matched that of *C. caudatus*. CD2-CD6 showed melting temperatures in the range of 79.90–80.00°C and were identified as *M*.

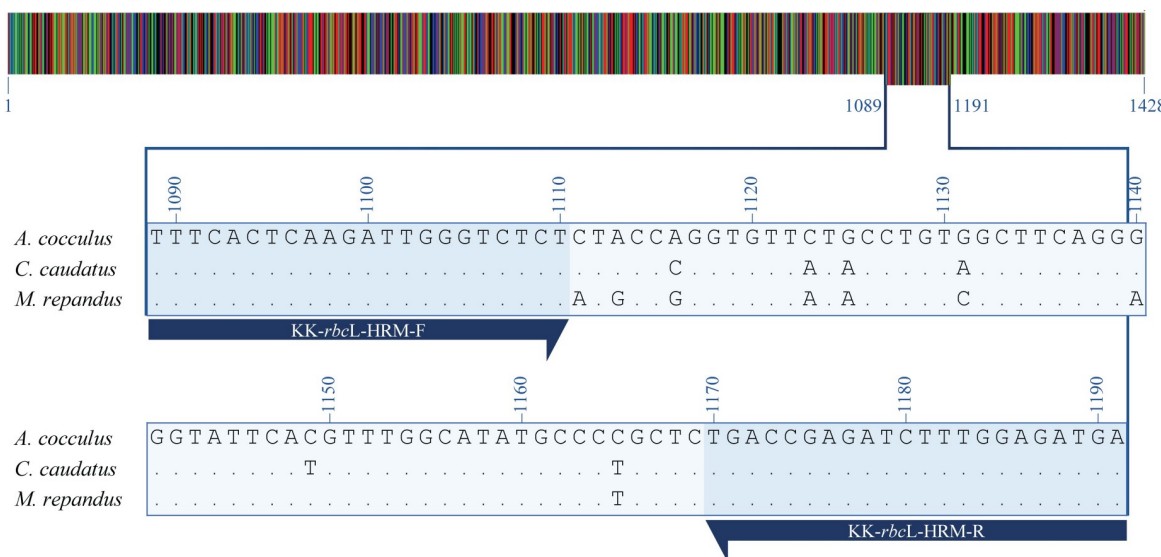

**Fig 4. Illustration of the *rbc*L target region for Bar-HRM analysis on the alignment of *A. cocculus*, *C. caudatus* and *M. repandus* with nucleotide polymorphic sites.** Blue arrows present forward primers (KK-rbcL-HRM-F) and reverse primers (KK-rbcL-HRM-R) with their directions. Consensus sequences are indicated with dots. The altered bases indicate sequence differences.

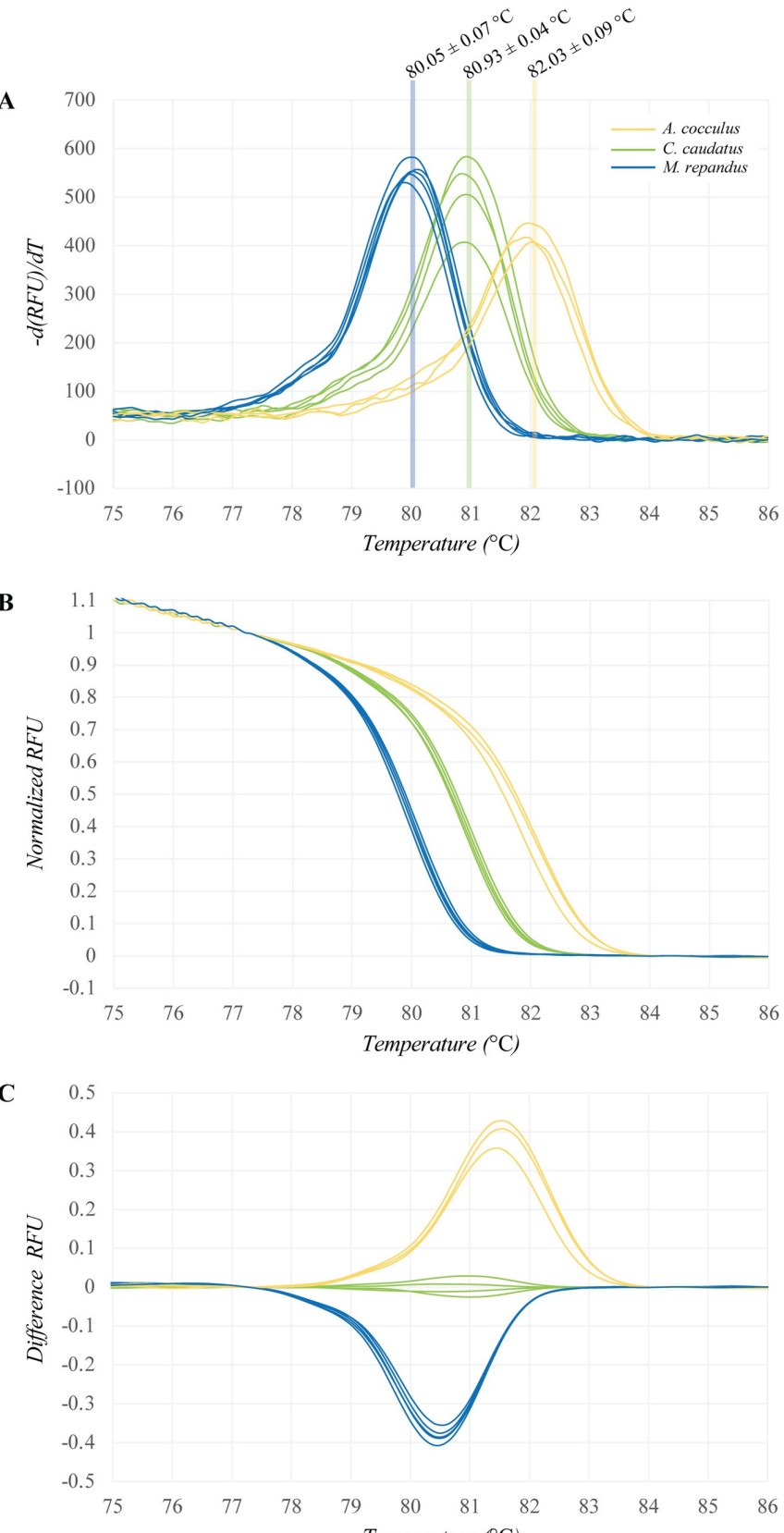

**Fig 5. High-resolution melting analysis using Bar-HRM primers targeting the *rbc*L region of *A. cocculus* (orange), *C. caudatus* (green) and *M. repandus* (blue).** (A) Melting curve plot presenting the melting temperature ($T_m$), (B) normalized plot and (C) difference plot.

*repandus* (Table 4). Distinct curve patterns for each mixture sample were obtained by Bar-HRM analysis (Fig 6). Difference plots of the two-species mixtures with various mixing ratios clearly separated the mixtures from the authentic species (Fig 6A–6D). Moreover, the three-species mixture sample was separated from the two-species mixtures in the difference plot (Fig 6E). Among the four laboratory-made YPSKK formulae, Bar-HRM analysis revealed overlapping patterns of difference plots from F-A, F-M, F-C and F-ACM, which made Bar-HRM unable to identify the species in the herbal formulae. In the laboratory-made YPSKK without any Kho-Khlan plants, Bar-HRM analysis showed different plots compared to those of F-A, F-M, F-C and F-ACM (Fig 6F).

## Discussion

Confusion of herbal materials due to the same vernacular name may impact consumer safety and treatment efficiency. Numerous reports have identified problems with the name used for medicinal plants. For example, *Pueraria candollei* Wall. ex Benth., *Butea superba* Roxb. ex Willd. and *Mucuna collettii* Lace are all called "Kwao Khruea" in Thai. However, misidentification of the Kwao Khruea species may lead to undesirable effects because the species have different properties [26]. Two popular vegetables, *Melientha suavis* Pierre and *Sauropus androgynus* (L.) Merr. share a common name, "Phak Wan", with the poisonous species *Urobotrya siamensis* Hiepko [24]. Unintentional consumption of *U. siamensis* resulted in comas and deaths in 2005 [27].

Recently, a number of reports have been published on the successful application of Bar-HRM and HPTLC analysis for the identification of related and nonclosely related species in herbal medicines. In 2018, Dual et al. applied the Bar-HRM method to identify Rhizoma Paridis and its common adulterants [28]. *Acanthus ebracteatus* Vahl, *Andrographis paniculate* (Burm.f.) Nees and *Rhinacanthus nasutus* (L.) Kurz were successfully discriminated by Bar-HRM analysis [29]. Moreover, Bar-HRM was applied to differentiate the poisonous plant *U. siamensis* from the edible vegetables *M. suavis* and *S. androgynus* for consumer safety purposes [24]. The HPTLC fingerprint revealed different phytochemical profiles between two nonrelated species, *Cyanthillium cinereum* (L.) H. Rob. (a smoking cessation herb) and its

**Table 4. Bar-HRM analysis showing the $T_m$ (˚C) of authentic plant species and purchased crude drug samples.**

| Samples | $T_m$ (˚C) | Claimed species | Detected species |
|---|---|---|---|
| **Authentic species** | | | |
| *A. cocculus* | 82.03±0.09 | - | *A. cocculus* |
| *C. caudatus* | 80.93±0.04 | - | *C. caudatus* |
| *M. repandus* | 80.05±0.07 | - | *M. repandus* |
| **Crude drug samples** | | | |
| CD1 | 80.84±0.06 | *M. repandus* | *C. caudatus* |
| CD2 | 79.90±0.07 | *M. repandus* | *M. repandus* |
| CD3 | 79.93±0.04 | *M. repandus* | *M. repandus* |
| CD4 | 79.93±0.04 | *M. repandus* | *M. repandus* |
| CD5 | 80.00±0.04 | *M. repandus* | *M. repandus* |
| CD6 | 79.90±0.07 | *M. repandus* | *M. repandus* |

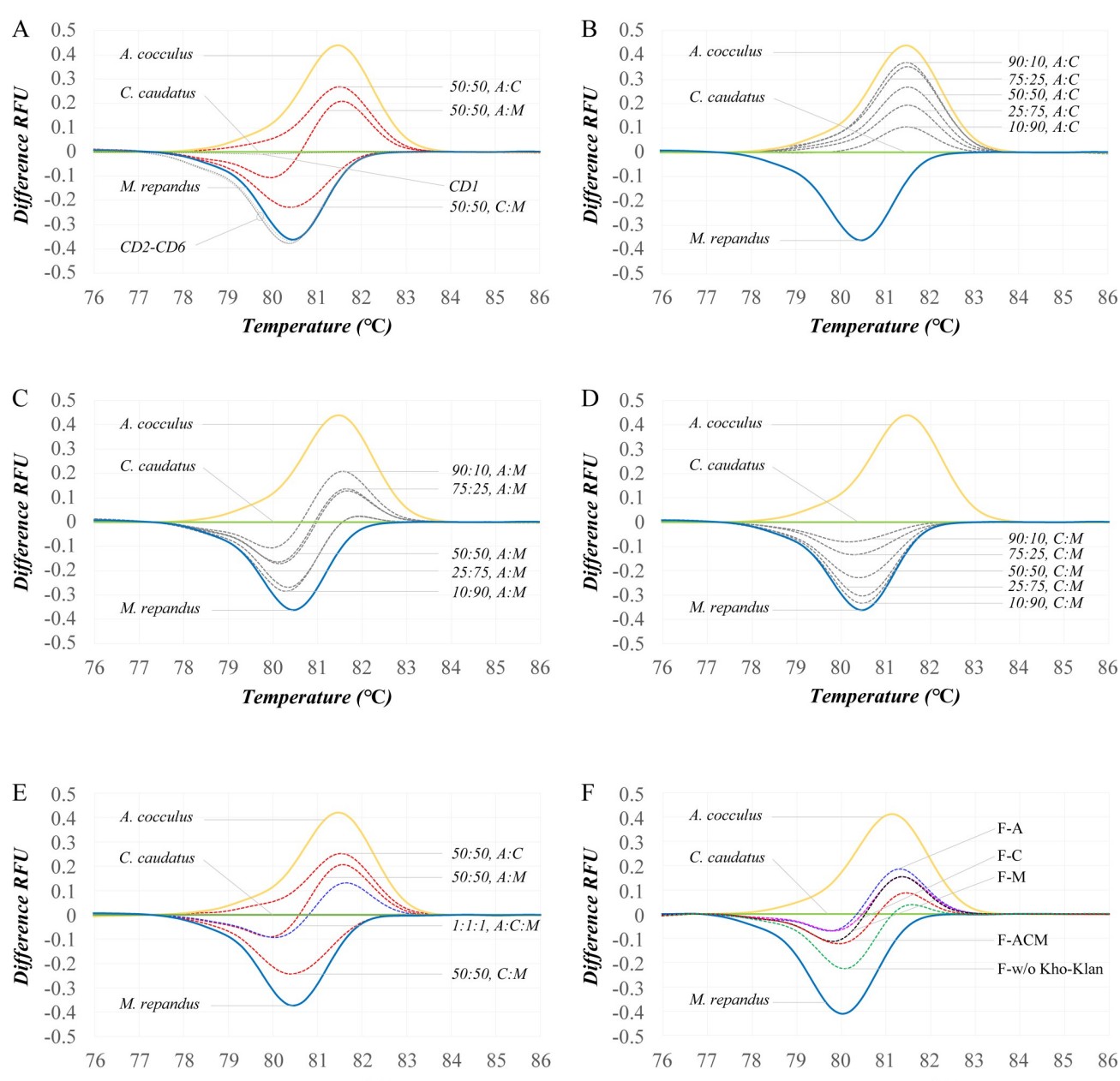

**Fig 6. Difference plots of samples obtained by Bar-HRM.** (A) Purchased Kho-Khlan crude drugs, (B) *A. cocculus* and *C. caudatus* mixture, (C) *A. cocculus* and *M. repandus* mixture, (D) *C. caudatus* and *M. repandus* mixture, (E) mixture of two and three species at equal amounts, and (F) laboratory-made YPSKK formulae. Mixture ratios are indicated. Authentic *A. cocculus*, *C. caudatus* and *M. repandus* are included.

adulterant, *Emilia sonchifolia* (L.) DC [19]. Combining HPTLC with the DNA barcode technique was previously reported for the identification of herbal raw materials such as *Aristolochia* species [30].

Confusion in the use of *C. caudatus* or *A. cocculus* instead of *M. repandus* in a pain relief formula, YPSKK, would lead to treatment failure and impact consumer safety. Therefore, the HPTLC method as a phytochemical fingerprint was used for the differentiation of Kho-Khlan (*M. repandus*) from *C. caudatus* and *A. cocculus*. Application of the HPTLC fingerprinting

method for testing the purchased Kho-Khlan crude drugs revealed that the chemical constituents varied among samples, although equal amounts of plant materials were used in this study. Variation in chemical composition may be influenced by environmental factors such as growth conditions, collection location, the plant part used and plant age [31]. Although the HPTLC profiles of *M. repandus*, *C. caudatus* and *A. cocculus* fluctuated, some HPTLC bands specific to each species were observed in the polyherbal mixture samples. This result validates the performance of the HPTLC technique for the identification of multiherbal formulae. This result agrees with reports showing that the HPTLC method is able to identify species within herbal formulae, such as an Iranian traditional medicine formula called "Zemad" and a multi-herbal ingredient formula called "Gegen Qinlian decoction" [32,33].

Bar-HRM analysis is a versatile, sequencing-free and reliable method. The assay has proven accurate in the rapid identification of species in diverse research fields, for instance, herbal medicine and their commercial products [34], medicines [35] and food science [36]. However, the suitable location for Bar-HRM primers should be carefully considered. From the DNA barcode sequence analysis results, the ITS region exhibited a higher percentage of nucleotide variation than the other candidate regions, which resulted in high variation and rendered the ITS a worse choice for Bar-HRM primers, similar to the results for the *psb*A-*trn*H intergenic spacer region. The *mat*K gene has been reported to have high discrimination power for species identification [24]. In our study, however, nucleotide sequences in the *mat*K gene of the three plants from different genera were variable and caused this region to be unsuitable for designing Bar-HRM primers. Since the gene sequences in the *rbc*L regions of *A. cocculus*, *C. caudatus* and *M. repandus* possess two conserved sites flanking nine nucleotide polymorphism sites, this region is suitable for the design of Bar-HRM primers. The *rbc*L region was chosen as a targeted amplified region for Bar-HRM analysis.

The nucleotide variation within 102 bp of PCR amplicons amplified from the three species resulted in different melting temperatures when analyzed by the Bar-HRM approach. An amplicon of 102 bp is in the range of desired amplicon lengths for the Bar-HRM analysis (<300 bp) suggested by Osathanunkul et al., 2015 [25]. The melting temperature obtained by Bar-HRM analysis remained unchanged in the fourth round of DNA template dilution (S3 Appendix). This finding was consistent with previous works on the stability of HRM results showing that the melting temperature did not vary within four logarithms of the initial concentration [23,37]. Moreover, the use of the *rbc*L region for species differentiation at the genus level has been revealed [38]. These results support our conclusion on the reliability of the *rbc*L region as a potential DNA barcode marker for discrimination of the nonrelated species that belong to different genera, *A. cocculus*, *C. caudatus* and *M. repandus*. The results from Bar-HRM analysis were obtained within 3.5 h, which shortened the detection time compared to that of Sanger sequencing, the gold standard.

Application of Bar-HRM analysis for testing claimed Kho-Khlan crude drugs (CD1-CD6) revealed that five (CD2, 3, 4, 5 and 6) out of six crude drugs were *M. repandus*, the correct species for preparing YPSKK formulae, which was confirmed by sequencing data (S1 Fig). Although the crude drug CD1 exhibited an ambiguous phytochemical pattern in the HPTLC assay, Bar-HRM analysis yielded a $T_m$ of 80.84±0.06°C, indicating that it was *C. caudatus*, not *M. repandus*, as claimed. This suggested that Bar-HRM and HPTLC can complement each other to distinguish *C. caudatus* and *M. repandus* when uncertainty in phytochemical constituents is observed. Bar-HRM analysis using genetic information can be used to clarify the ambiguous result, as the genetic information is stable. The poison species *A. cocculus* was fortunately not detected in any crude drugs. The detection of *C. caudatus* implies that drugs with incorrect species labels are sold on the market; therefore, more attention should be given to quality control in terms of the identification of Kho-Khlan crude drugs. In the present study, Bar-HRM

analysis revealed the specificity of normalized and difference plots to DNA ratios of two- and three-species mixtures, which should be further developed for quantitative detection in the future. However, Bar-HRM may be limited for the identification of polyherbal formulae; therefore, more sensitive DNA methods, such as next-generation sequencing (NGS), could be applied. Taken together, this work suggests that Bar-HRM is a practical approach for the identification of raw materials and can complement the HPTLC method when phytochemical profiles exhibit unclear results and vice versa.

## Conclusion

YPSKK is a multiherbal formula for pain relief treatment in the NLEM of Thailand. The main ingredient, *M. repandus*, shares the vernacular name Kho-Khlan with *C. cocculus* and *M. caudatus*. This can cause confusion in terms of usage and may have serious effects via either toxicity or unsuccessful treatment. The present study established a combined Bar-HRM and HPTLC technique for identifying the correct Kho-Khlan species, *M. repandus*. This method was successfully applied to identify crude drugs and multiherbal mixed formulae and serves as a quality control tool for preventing accidental confusion of herbal species sharing the same common name. The DNA and chemical signatures of *M. repandus* obtained here can help manufacturers increase the quality control of *M. repandus* raw material in commercialized pain relief products.

## Supporting information

**S1 Fig. Confirmation of CD3 (*M. repandus*) by sequencing of PCR amplicons after Bar-HRM analysis.** (A) DNA alignment of the CD3 sequence with sequences of authentic *A. cocculus*, *C. caudatus* and *M. repandus*, (B) electropherogram showing the partial sequence obtained from the Bar-HRM amplicon. The red box presents identical nucleotide sequences among the authentic *M. repandus* sequence and CD3 sequence. The green box shows the same area of nucleotide alignment and electropherogram. "·" indicates an identical nucleotide sequence. "-" indicates no electropherogram result.
(PDF)

**S1 Appendix. Additional primers used for DNA barcode generation in this study.**
(PDF)

**S2 Appendix. Sequence alignment of four core DNA barcode regions among *A. cocculus*, *C. caudatus* and *M. repandus*.**
(PDF)

**S3 Appendix. Melting temperatures of PCR amplicons and cluster groups generated by various DNA concentrations.**
(PDF)

**S1 Raw images.**
(PDF)

## Acknowledgments

This research was supported by the Ratchadapisek Somphot Fund for Postdoctoral Fellowship, Chulalongkorn University. The authors are grateful to Professor Chaiyo Chaichantippayut for species identification. We thank the Faculty of Pharmaceutical Sciences, Chulalongkorn University, for providing facilities.

## Author Contributions

**Conceptualization:** Kannika Thongkhao, Suchada Sukrong.

**Data curation:** Kannika Thongkhao, Vipawee Pichetkun, Suthathip Gaewtongliam.

**Formal analysis:** Kannika Thongkhao.

**Investigation:** Kannika Thongkhao, Chayapol Tungphatthong.

**Methodology:** Kannika Thongkhao.

**Project administration:** Suchada Sukrong.

**Resources:** Vipawee Pichetkun, Suthathip Gaewtongliam, Worakorn Wiwatcharakornkul.

**Supervision:** Suchada Sukrong.

**Visualization:** Chayapol Tungphatthong.

**Writing – original draft:** Kannika Thongkhao.

**Writing – review & editing:** Suchada Sukrong.

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
