## [Decision Letter · Decision Letter 0]

7 Mar 2022

PONE-D-22-00460Combining DNA and HPTLC
profiles to differentiate a pain relief herb, Mallotus repandus, from plants sharing
the same common name (“Kho-Khlan”), Anamirta cocculus and Croton
caudatusPLOS ONE

Dear Dr. Sukrong,

Thank you for submitting your manuscript to PLOS ONE. After careful consideration, we
feel that it has merit but does not fully meet PLOS ONE’s publication criteria as it
currently stands. Therefore, we invite you to submit a revised version of the
manuscript that addresses the points raised during the review process.

Introduction should be more focused.

Additional clarification related to material and methodology used are needed.

Table 2 should be reorganized as suggested by Reviewer #1. In Figure 3 some data are
missing.

The authors should considerably widen the Discussion section towards providing
scientific and empirical justification of the presented methodologies for
non-closely related species.

The manuscript would greatly benefit if being proofread by a native English
speaker.

Please submit your revised manuscript by Apr 21 2022 11:59PM. If you will need more
time than this to complete your revisions, please reply to this message or contact
the journal office at plosone@plos.org. When
you're ready to submit your revision, log on to https://www.editorialmanager.com/pone/ and select the 'Submissions
Needing Revision' folder to locate your manuscript file.

Please include the following items when submitting your revised
manuscript:A rebuttal letter that responds to each point raised by the academic
editor and reviewer(s). You should upload this letter as a separate file
labeled 'Response to Reviewers'.A marked-up copy of your manuscript that highlights changes made to the
original version. You should upload this as a separate file labeled
'Revised Manuscript with Track Changes'.An unmarked version of your revised paper without tracked changes. You
should upload this as a separate file labeled 'Manuscript'.

If you would like to make changes to your financial disclosure, please include your
updated statement in your cover letter. Guidelines for resubmitting your figure
files are available below the reviewer comments at the end of this letter.

We look forward to receiving your revised manuscript.

Kind regards,

Branislav T. Šiler, Ph.D.

Academic Editor

PLOS ONE

Journal Requirements:

(This research was supported by the Ratchadapisek Somphot Fund for Posdoctoral
Fellowship, Chulalongkorn University. The authors are grateful to Professor Chaiyo
Chaichantippayut for species identification. We thank the Faculty of Pharmaceutical
Sciences, Chulalongkorn University, for providing facilities.)

(This research was supported by the Ratchadapisek Somphot Fund for Posdoctoral
Fellowship, Chulalongkorn University. The authors are grateful to Professor Chaiyo
Chaichantippayut for species identification. We thank the Faculty of Pharmaceutical
Sciences, Chulalongkorn University, for providing facilities.)

(The authors have declared that no competing interests exist.)

Please include your amended statements within your cover letter; we will change the
online submission form on your behalf."

Additional Editor Comments:

Introduction should be more focused.

Additional clarification related to material and methodology used are needed.

Table 2 should be reorganized as suggested by Reviewer #1. In Figure 3 some data are
missing.

The authors should considerably widen the Discussion section towards providing
scientific and empirical justification of the presented methodologies for
non-closely related species.

The manuscript would greatly benefit if being proofread by a native English
speaker.

Reviewers' comments:

Reviewer's Responses to Questions

**Comments to the Author**

1. Is the manuscript technically sound, and do the data support the conclusions?

Reviewer #1: Partly

Reviewer #2: Yes

2. Has the statistical analysis been performed
appropriately and rigorously? 

Reviewer #1: No

Reviewer #2: Yes

3. Have the authors made all data underlying the
findings in their manuscript fully available?

Reviewer #1: Yes

Reviewer #2: Yes

4. Is the manuscript presented in an intelligible
fashion and written in standard English?

Reviewer #1: No

Reviewer #2: Yes

5. Review Comments to the Author

Reviewer #1: The research article entitled “Combining DNA and HPTLC profiles to
differentiate a pain relief herb, Mallotus repandus, from plants sharing the same
common name (“Kho-Khlan”), Anamirta

cocculus and Croton caudatus” highlights the use of HPTLC and BAR-HRM to distinguish
the study plants. The study is potentially interesting, it has to be improved a lot
before it is suitable for publication.

Comments

• One of the foremost concerns is that though the article is well written, still the
proficiency of English is lacking in the article and it needs to improved.

• Introduction (line 93-94): “Phytochemical composition may be uncertain due to
environmental factors”. What authors would like to convey by making such statements
in the introduction? This sentence is presented out of nowhere in introduction.

• Line 101: What are those benefits and limitations of BAR-HRM, and what are the
benefits of using integrative approach for identification of plants?

• The introduction lacks a coherence, and authors failed to showcase why they wanted
to use HPTLC and BAR-HRM in their study, why not simple TLC or other chromatographic
methods to distinguish the study species.

• Methodology: Authors failed to provide the details of herbarium voucher specimen
details for the study species?. Also, authors state that fresh leaves were
collected, however, they used stem samples for HPTLC analysis? Therefore, authors
needs to specify the source of such stem samples used in HPTLC.

• Line 143: Crushed? How did the authors crushed the dried stem samples?

• Line 147: sprayed or spotted?

• Line 150: Authors state that the track distance is 11.4 mm, however they did not
indicate whether it is from lower edge of the plate or from the sample spot?

• Table 2: It is confusing to see these primer pairs and sequence in table 2. Authors
indicate in few primers as “This study”, then what is the purpose of providing other
primer pair details in table 2? Also, ““This study”, is not indicated in ITS and
psbA-trnH primer pairs. Authors should clarify and include only the primer pairs
used in the study, also if they used more than one primer pair to amplify one
particular barcode region, it should be included in discussion part.

• Authors failed to provide the details of primer design and the tools used to design
for the BAR-HRM primer “KK-rbcL-HRM-F” and “KK-rbcL-HRM-R”

• Authors can really shorten the DNA extraction part and PCR amplification part in
the methodology section

• One of my concern with the DNA barcoding and BAR-HRM in this study is, authors have
not used any closely related plant groups as control or to highlight that BAR-HRM
can distinguish closely related species. It is obvious and no surprise that the
study plants belong to Euphorbiaceae and Menispermaceae which shows enough genetic
variation to be distinguished. In fact, experienced filed botanist and microscopy
analysis can easily distinguish these study species, why anyone wants to use BAR-HRM
to distinguish plants that belong to different families and has enough morphological
variation itself.

• Another concern with HPTLC is, why authors have not derivatized the HPTLC plate
using any derivatizing reagents?, without derivatization, and using only 366 nm,
only UV active compounds are visible and the data provided by authors are not
sufficient to claim that the authors developed a HPTLC method.

Reviewer #2: The present study established a combined Bar-HRM and HPTLC technique to
identify the correct species, Mallotus repandus, from authentic Anamirta cocculus
and Croton caudatus. This method was successfully applied to identify crude drugs
and multiherbal mixed formulae and serve as a quality control tool for preventing
accidental confusion of herbal species sharing the same common name. The experiment
design is logical and fine. However, there are still some issues that merit authors’
attention.

1. The writing of the “Introduction” is too wordy, especially lines 59-76. Simply
explain that A. cocculus and C. caudatus can't be used as substitutes of M. repandus
in clinical practices.

2. The description of the existing identification methods in the second paragraph of
“Introduction” is logically confused. Please rewrite.

3. Why did authors select the rbcL region for bar-HRM analysis for the
differentiation of M. repandus from A. cocculus and C. caudatus?

4. I cannot find the HPTLC bands of C. caudatus (Rf = 0.55) in tracks 6-7 in figure
3. Besides, the enlarged image on the right in Figure 3 should have a scale.

5. This study showed that compared with A. cocculus, species C. caudatus and M.
repandus were more difficult to distinguish. It is suggested that authors add some
discussion to elaborate on how the two methods, Bar-HRM and HPTLC, complement each
other to distinguish the three species, especially the distinction between C.
caudatus and M. repandus.

6. PLOS authors have the option to publish the peer
review history of their article (what does this mean?). If published, this will
include your full peer review and any attached files.

If you choose “no”, your identity will remain anonymous but your review may still be
made public.

**Do you want your identity to be public for this peer review?** For
information about this choice, including consent withdrawal, please see our
Privacy Policy.

Reviewer #1: **Yes: **Seethapathy Gopalakrishnan Saroja

Reviewer #2: No

---

## [Author Response · Author response to Decision Letter 0]

22 Apr 2022

We appreciate the editor and the reviewers for the constructive comments. Each
comment has been carefully considered point by point and responded. Responses to the
reviewers and changes in the revised manuscript are as following:

A point-by-point response to the comments

Editor comments

Comment 1: Thank you for submitting your manuscript to PLOS ONE. After careful
consideration, we feel that it has merit but does not fully meet PLOS ONE’s
publication criteria as it currently stands. Therefore, we invite you to submit a
revised version of the manuscript that addresses the points raised during the review
process.

Response: Thank you editor for providing us a change to improve the manuscript. You
encouraged us to revise as best as we can to make the manuscript high quality to
publish in PLOS ONE.

Comment 2: Introduction should be more focused.

Response: The introduction section was revised to be more focused as suggestion from
editor and reviewer #1 and #2. Please read our responses to “Comment 4 of the
reviewer #1” and “Comment 1 of the reviewer #2”. 

Comment 3: Additional clarification related to material and methodology used are
needed.

Response: The manuscript has been revised. Additional clarification related to
material and methodology used in the study has been added in the “Introduction” and
“Materials and Methods” sections. 

“Introduction” Section

Line 84-97 and Line 100-113 were added to the Introduction Section to provide more
information to the HPTLC and Bar-HRM methods, respectively, as following:

Line 84-97 of the revised manuscript: “Thin-layer chromatography (TLC) and
high-performance TLC (HPTLC), which are recommended in the herbal pharmacopoeias of
many countries, including Thailand, are reliable methods for phytochemical
constituent examination; however, the methods require a target compound as a
standard reference [17, 18]. HPTLC, a sophisticated form of TLC, provides good
separation efficiency due to the higher quality of its separation plate. HPTLC
exhibits higher accuracy, reproducibility, and ability to document the results than
TLC [18]. Therefore, this method has been used to determine the phytochemical
profile of herbal species. However, uncertain results may occur due to environmental
factors that affect the chemical composition of herbal species and biological
activities of the substances [19]. In recent years, a molecular approach called the
DNA barcoding technique has gained demand in species identification because it is an
accurate, cost-effective and reliable tool for species identification. The DNA
barcoding method provides species-level information, and small amounts of samples
are needed for the identification process [19].”

Line 100-113 of the revised manuscript: Bar-HRM, a sequencing free method, detects
signal alteration during the dissociation of double-stranded DNA into single
stranded DNA. Each plant species can be differentiated by their individual melting
temperature (Tm) which correlated to their nucleotide sequences in the target region
[23]. Bar-HRM analysis exhibits fast, cost-effective and reliable method, moreover,
small amount of sample is required for species identification. However, Bar-HRM
primer design is challenging when the target sequence has high variation rates
across the target amplicon and the Bar-HRM analysis has limited when low quality of
DNA template is used [24]. As mentioned above, each identification method has
advantages and limitations, therefore, an integrative approach is proposed to
differentiate substitutions and adulterants of herbal species [19, 25]. Combined
methods can be applied to prevent the use of incorrect herbal species and support
the quality of herbal materials to meet international standards [19].

“Materials and Methods” Section The section has been revised. 

Original:

Line 121-123 of the original manuscript: Fresh leaves of A. cocculus (n=8), C.
caudatus (n=8) and M. repandus (n=8) were collected from various locations across
Thailand for the DNA barcoding experiment (Table 1).

Revision:

Line 121-123 of the revised manuscript: Fresh leaves and stems of A. cocculus (n=8),
C. caudatus (n=8) and M. repandus (n=8) were collected from various locations across
Thailand (Table 1). These collections are legally permitted.

More information was added in the revised manuscript.

Line 129-130 of the revised manuscript: All experiments were performed in accordance
with relevant guidelines and regulations.

Comment 4: Table 2 should be reorganized as suggested by Reviewer #1.

Response: Table 2 has been reorganized as suggested by Reviewer #1. Selected
sequencing primers were removed to Appendix S1. Please see the "Response to
Reviewers" file.

Comment 5: In Figure 3 some data are missing.

Response: Thank you editor for asking. The authors repeated the HPTLC experiments in
order to see the missing species-specific band of C. caudatus. We adjusted the
mobile phase from toluene:acetone:formic acid mixture (5:3:0.5, v/v/v) into
toluene:acetone:formic acid mixture (5:4:0.5, v/v/v) . This adjustment of mobile
phase system affects the Rf in the HPTLC result therefore, the HPTLC result was
rewrite. The original Fig. 2 and Fig. 3 were replaced by the revised Fig. 2 and the
revised Fig. 3, respectively. Please see the revised figures in the "Response to
Reviewers" file.

Additional rewrite after changing of mobile phase in HPTLC experiments.

Line 167 of the original manuscript: 

Toluene:acetone:formic acid (5:-:0.5, v/v/v) was used as the mobile phase

Line 167 of the revised manuscript: Toluene:acetone:formic acid (5:4:0.5, v/v/v) was
used as the mobile phase.

The HPTLC results were revised as following.

Original sentences: 

Line 229-230 of the original manuscript: “Species-specific bands were obtained for
authentic A. cocculus (Rf = 0.18), C. caudatus (Rf = 0.08 and 0.55) and M. repandus
species (Rf = 0.05, 0.20, 0.63 and 0.68).”

Line 232-237 of the original manuscript: “A bright blue band at Rf = 0.68 for M.
repandus was found in CD2-CD6. The blue band at Rf = 0.05 was present in CD2-CD5,
while the band at Rf = 0.20 was present in CD2 and CD3. The crude drug CD1 showed an
ambiguous HPTLC pattern with a faint band at Rf = 0.08. No distinct bands of A.
cocculus (Rf = 0.18) or C. caudatus (Rf = 0.55) were detected in any of the crude
drug samples (Fig 2).”

Revised sentences:

Line 245-246 of the revised manuscript: “Species-specific bands were obtained from
authentic A. cocculus (Rf = 0.22), C. caudatus (Rf = 0.02 and 0.60) and M. repandus
species (Rf = 0.08, 0.26, 0.68 and 0.72).”

Line 248-253 of the revised manuscript: “A bright blue band at Rf = 0.72 for M.
repandus was found in CD2-CD6. The blue band at Rf = 0.08 was present in CD2-CD6,
while the band at Rf = 0.26 was present in CD2-CD6. The crude drug CD1 showed an
ambiguous HPTLC pattern with a faint band at Rf = 0.02. No distinct bands of A.
cocculus (Rf = 0.22) or C. caudatus (Rf = 0.60) were detected in any of the crude
drug samples (Fig 2).”

Comment 6: The authors should considerably widen the Discussion section towards
providing scientific and empirical justification of the presented methodologies for
non-closely related species.

Response: The manuscript has been revised by widen the Discussion section as
recommended by the editor. The authors added a paragraph according to the scientific
and empirical justification of the presented methodologies for non-closely related
species in the discussion section as following. 

Line 353-364 of the revised manuscript: “Recently, a number of reports have been
published on the successful application of Bar-HRM and HPTLC analysis for the
identification of related and nonclosely related species in herbal medicines. In
2018, Dual et al. applied the Bar-HRM method to identify Rhizoma Paridis and its
common adulterants [28]. Acanthus ebracteatus Vahl, Andrographis paniculate
(Burm.f.) Nees and Rhinacanthus nasutus (L.) Kurz were successfully discriminated by
Bar-HRM analysis [29]. Moreover, Bar-HRM was applied to differentiate the poisonous
plant U. siamensis from the edible vegetables M. suavis and S. androgynus for
consumer safety purposes [24]. The HPTLC fingerprint revealed different
phytochemical profiles between two nonrelated species, Cyanthillium cinereum (L.) H.
Rob. (a smoking cessation herb) and its adulterant, Emilia sonchifolia (L.) DC.
[19]. Combining HPTLC with the DNA barcode technique was previously reported for the
identification of herbal raw materials such as Aristolochia species [30].”

Comment 7: The manuscript would greatly benefit if being proofread by a native
English speaker.

Response: For the revised version, the manuscript has been proofread and edited by
the American Journal Experts (AJE) service in order to improve the language. The AJE
certificated was attached. 

 

Reviewer #1: 

The research article entitled “Combining DNA and HPTLC profiles to differentiate a
pain relief herb, Mallotus repandus, from plants sharing the same common name
(“Kho-Khlan”), Anamirta cocculus and Croton caudatus” highlights the use of HPTLC
and BAR-HRM to distinguish the study plants. The study is potentially interesting,
it has to be improved a lot before it is suitable for publication.

Comments 1: One of the foremost concerns is that though the article is well written,
still the proficiency of English is lacking in the article and it needs to
improved.

Response: Thank you reviewer for your suggestions. The authors apologize for the
unproficiency of English in the manuscript. 

For the revised manuscript, the authors submitted the manuscript for a language
service in order to improve it English proficiency and the revised manuscript has
been edited by the American Journal Experts service (AJE). Please find the
certificate in the "Response to Reviewers" file.

Comments 2: Introduction (line 93-94): “Phytochemical composition may be uncertain
due to environmental factors”. What authors would like to convey by making such
statements in the introduction? This sentence is presented out of nowhere in
introduction.

Response: We are sorry about this mistake. Authors agree with the reviewer. The
sentence has been deleted. 

Comments 3: Line 101: What are those benefits and limitations of BAR-HRM, and what
are the benefits of using integrative approach for identification of plants?

Response: BAR-HRM analysis has benefits and limitations. The authors added more
information about benefits and limitations of BAR-HRM in line 100-108. The benefit
of using integrative approach for identification of plants is added in the line
102-105.

Revised sentences:

Line 100-108 of the revised manuscript: “Bar-HRM, a sequencing-free method, detects
signal alterations during the dissociation of double-stranded DNA generated from the
PCR into single-stranded DNA. Each plant species can be differentiated by their
individual melting temperature (Tm), which is correlated to their nucleotide
sequences in the target region [23]. Bar-HRM analysis is a fast, cost-effective and
reliable method; moreover, a small amount of sample is required for species
identification. However, Bar-HRM primer design is challenging when the target
sequence has high variation rates across the target amplicon, and Bar-HRM analysis
is limited when low-quality DNA templates are used [24].”

Line 109-113 of the revised manuscript: “As mentioned above, each identification
method has advantages and limitations; therefore, an integrative approach is
proposed to differentiate substitutions or adulterants of herbal species [19, 25].
Combined phytochemical profiles and DNA information can be applied to prevent the
use of incorrect herbal species and support the quality of herbal materials to meet
international standards [19].”

Comments 4: The introduction lacks a coherence, and authors failed to showcase why
they wanted to use HPTLC and BAR-HRM in their study, why not simple TLC or other
chromatographic methods to distinguish the study species.

Response: Thank you for bringing this into notice. 

In this study, the authors designed to use HPTLC instead of the TLC method as the
HPTLC has higher resolution than that of the simple TLC, therefore, the HPTLC was
chosen to distinguish phytochemical pattern in this work. Therefore, the author
revised the manuscript by adding the advantage of HPTLC in Line 85-93. As the
introduction part lacks a coherence, the manuscript has been edited to make a
coherence within the section by adding more information in the introduction part to
explain the reason of using the HPTLC and BAR-HRM method for species identification
in this study in Line 85-93 and Line 100-108, respectively.

Line 85-93 of the revised manuscript: “Thin-layer chromatography (TLC) and
high-performance TLC (HPTLC), which recommended in herbal pharmacopoeias of many
countries, are reliable methods for phytochemical constituents examination because
the methods require target compound as standard reference [17, 18]. HPTLC, a
sophisticated form of TLC, provides good separation efficiency due to higher quality
of its separation plate. HPTLC exhibits higher accuracy, reproducibility, and
ability to document the results compare to TLC [18]. Therefore, this method has been
used for phytochemical profile of herbal species. However, uncertain results may
occur by environmental factors which affect the chemical composition of herbal
species and biological activities of the substances [19].”

Line 100-108 of the revised manuscript: “Bar-HRM, a sequencing free method, detects
signal alteration during the dissociation of double-stranded DNA into single
stranded DNA. Each plant species can be differentiated by their individual melting
temperature (Tm) which correlated to their nucleotide sequences in the target region
[23]. Bar-HRM analysis exhibits fast, cost-effective and reliable method, moreover,
small amount of sample is required for species identification. However, Bar-HRM
primer design is challenging when the target sequence has high variation rates
across the target amplicon and the Bar-HRM analysis has limited when low quality of
DNA template is used [24].”

Comments 5: Methodology: Authors failed to provide the details of herbarium voucher
specimen details for the study species? 

Response: In the previous manuscript version, the authors used the “Code” to present
herbarium voucher specimen in Table 1. Therefore, in this revised manuscript, the
authors change the column “Code” to “Voucher number”. 

Comments 6: Methodology: Also, authors state that fresh leaves were collected,
however, they used stem samples for HPTLC analysis? Therefore, authors need to
specify the source of such stem samples used in HPTLC.

Response: Thank you reviewer for bringing into notice. In this study, stems were used
for HPTLC analysis. Therefore, the revised version of manuscript was edited in Line
121-123 to specify source of sample used in HPTLC experiment. The authors also
revised the Method part (HPTLC profiles) and provided detail of sample (Line 156-157
in revised version) used in HPTLC analysis as following. 

Original sentence: 

Line 110-111 of the original manuscript: “Fresh leaves of A. cocculus (n=8), C.
caudatus (n=8) and M. repandus (n=8) were collected from various locations across
Thailand for the DNA barcoding experiment (Table 1).”

Revised sentence: 

Line 121-123 of the revised manuscript: “Fresh leaves and stems of A. cocculus (n=8),
C. caudatus (n=8) and M. repandus (n=8) were collected from various locations across
Thailand (Table 1). These collections are legally permitted.”

Original sentence: 

Line 142-143 of the original manuscript: “To obtain the phytochemical profiles of A.
cocculus, C. caudatus and M. repandus, 1 g of dried stems from each species were
crushed into a fine powder.”

Revised sentence: 

Line 156-157 of the revised manuscript: “To obtain the phytochemical profiles of
selected samples, including A. cocculus (SS-628), C. caudatus (SS-537) and M.
repandus (SS-583),…”

Comments 7: Line 143: Crushed? How did the authors crushed the dried stem
samples?

Response: We are sorry for missing information. The author crushed the dried stem
samples using a grinder. The sentence has been revised in line 157-158.

Original sentence: 

Line 142-143 of the original manuscript: “…,1 g of dried stems from each species were
crushed into a fine powder.”

Revised sentence: 

Line 157-158 of the revised manuscript: “…,1 g of dried stems from each species was
crushed into a fine powder using a M 20 Universal mill grinder (IKA, Germany).”

Comments 8: Line 147: sprayed or spotted? 

Response: Sorry for the mistake. The sentence has been edited by changing “sprayed to
spotted” in Line 162-163.

Original sentence: 

Line 147 of the original manuscript: “Then, 5 µl of the extracted solution was
sprayed onto an HPTLC plate…”

Revised sentence: 

Line 162-163 of the revised manuscript: “Then, 5 µl of the extracted solution was
spotted onto an HPTLC plate...”

Comments 9: Line 150: Authors state that the track distance is 11.4 mm, however they
did not indicate whether it is from lower edge of the plate or from the sample
spot?

Response: Sorry for missing information. The sentence has been revised by adding
details of the HPTLC plate setting in Line 164-166 of the revised manuscript.

Original manuscript: 

Line 149-150 of the original manuscript: “Each individual band was 8 mm in length,
the distance between tracks was 2 mm, and the track distance was 11.4 mm.” 

Revised manuscript: 

Line 164-166 of the revised manuscript: “Each individual band was 8 mm in length, the
distance between tracks was 2 mm, and the track distance was 11.4 mm from the lower
edge of the plate.”

Comments 10: Table 2: It is confusing to see these primer pairs and sequence in table
2. Authors indicate in few primers as “This study”, then what is the purpose of
providing other primer pair details in table 2? Also, ““This study”, is not
indicated in ITS and psbA-trnH primer pairs. Authors should clarify and include only
the primer pairs used in the study, also if they used more than one primer pair to
amplify one particular barcode region, it should be included in discussion part.

Response: Primers listed in Table 2 are the amplification and sequencing primers. The
“This study” terms refer to primers that are originally designed in this work. For
those primers without the term “This study” refer to primers that are previously
reported. This work, the DNA barcode regions; ITS and psbA-trnH intergenic spacers
regions, were amplifiable using the published ITS primers (ITS1 and ITS4) and
psbA-trnH primers (psbA-trnHF and psbA-trnHR), respectively. However, matK region
required more primers to complete the full length of nucleotide sequences for
Mallotus repandus and Croton caudatus. 

 As suggested by the reviewer, the authors agreed to reorganize the Table 2 and the
additional primers for matK region of Mallotus repandus and Croton caudatus have
been moved to the Appendix S1. Please see the detail in "Response to Reviewers"
file.

Comments 11: Authors failed to provide the details of primer design and the tools
used to design for the BAR-HRM primer “KK-rbcL-HRM-F” and “KK-rbcL-HRM-R”

Response: Thank you for raising to this point. The authors used the MUSCLE program
was used for DNA barcode sequences alignment followed by Primer 3 and BLAST software
for Bar-HRM primers design. 

The manuscript has been revised by adding details of primer design including tools
which used for the BAR-HRM primer “KK-rbcL-HRM-F” and “KK-rbcL-HRM-R” design. The
information has been added in line 210-212 and line 214-217. 

Line 210-212 of the revised manuscript: “To design Bar-HRM primers, nucleotide
sequences obtained from the four DNA barcode regions of M. repandus, A. cocculus and
C. caudatus were aligned by MUSCLE with gap open = –400; gap extend = 0; clustering
method = UPGMB; and Min Diag Length = 24.” 

Line 214-217 of the revised manuscript: “Primer 3 and BLAST software were used for
primer design. The Bar-HRM forward (KK-rbcL-HRM-F) and reverse primers
(KK-rbcL-HRM-R) were designed based on the conserved regions of the rbcL gene of the
three plants. The targeted amplicon provided a 102 bp amplicon with 9 polymorphic
sites of the rbcL gene.”

Comments 12: Authors can really shorten the DNA extraction part and PCR amplification
part in the methodology section.

Response: Thank you reviewer for your suggestion. The manuscript has been revised by
shorten the DNA extraction part and PCR amplification part in Line 175-184 of the
revised manuscript.

Original paragraph of DNA extraction part: 

Line 159-173 of the original manuscript: “Fresh leaves of authentic A. cocculus, C.
caudatus and M. repandus were ground into a fine powder with liquid nitrogen.
Genomic DNA was extracted from 50 mg of fine powder using a DNeasy Plant Mini Kit
(Qiagen, Germany) following the manufacturer’s instructions. A GENECLEAN Kit (MP
Biomedicals, USA) was used to purify the genomic DNA according to the manufacturer’s
protocol. The quantity of the extracted DNA was determined spectrophotometrically
using a NanoDrop One UV-Vis Spectrophotometer (Thermo Scientific, USA). DNA quality
was observed by agarose gel electrophoresis. Genomic DNA was run on 0.8% (w/v)
agarose in 1X TAE gel containing 1X RedSafe nucleic acid staining solution (iNtRON
Biotechnology, USA) at 100 V for 30 min. The agarose gel was analyzed with a UVP
GelSolo (Analytik Jena GmbH, Germany) gel documentation system. Images were taken by
onboard VisionWorks software (Analytik Jena GmbH, Germany). Genomic DNA was stored
at -20 ℃ for further use. Genomic DNA was extracted and purified from the purchased
crude drugs called “Kho-Khlan”, mixed powder of plants and laboratory-made YPSKK
formulae using the methods described above. Genomic DNA quantification and
qualification were conducted as described for the authentic plant samples.”

Revised paragraph of DNA extraction part: 

Line 175-184 of the revised manuscript: “Genomic DNA from leaves of the samples, the
purchased crude drugs, mixed herbal powder and laboratory-made YPSKK formulae were
extracted using a DNeasy Plant Mini Kit (Qiagen, Germany) and further purified using
a GENECLEAN Kit (MP Biomedicals, USA) according to the manufacturer’s protocol. DNA
quantity and quality were determined using a NanoDrop One UV–Vis Spectrophotometer
(Thermo Scientific, USA) and agarose gel electrophoresis, respectively. Genomic DNA
was run on 0.8% (w/v) agarose in 1X TAE gel containing 1X RedSafe nucleic acid
staining solution (iNtRON Biotechnology, USA) at 100 V for 30 min. Agarose gel was
analyzed with a UVP GelSolo (Analytik Jena GmbH, Germany) gel documentation system,
and images were taken by onboard VisionWorks software (Analytik Jena GmbH, Germany).
Genomic DNA was stored at -20℃ for further use.”

Comments 13: One of my concern with the DNA barcoding and BAR-HRM in this study is,
authors have not used any closely related plant groups as control or to highlight
that BAR-HRM can distinguish closely related species. It is obvious and no surprise
that the study plants belong to Euphorbiaceae and Menispermaceae which shows enough
genetic variation to be distinguished. In fact, experienced filed botanist and
microscopy analysis can easily distinguish these study species, why anyone wants to
use BAR-HRM to distinguish plants that belong to different families and has enough
morphological variation itself.

Response: Thank you reviewer for your comment. We totally appreciate your concern.
The reason that the authors have not used any closely related species in this study
because the plants called “Kho-Khlan”, in fact, they are only three herbal species;
A. cocculus, C. caudatus and M. repandus found in Thai herbal markets. Therefore, we
have not used other plants in this study.

In the herbal markets, normally the herbal materials are sold in the processed forms
such as small pieces of stem and powder which its identities have been lost. This is
challenging us to identify small pieces of stem or herbal powder by morphological
analysis or microscopic examination. Bar-HRM analysis will benefit people who
involved with regulatory policy of herbal products and herbal industry as the
Bar-HRM analysis supports species identification of highly processed raw materials.
The Bar-HRM analysis also requires low amount of DNA sample and the method is able
to amplify fragmented DNA, which normally found in the DNA of highly processed
materials. 

Comments 14: Another concern with HPTLC is, why authors have not derivatized the
HPTLC plate using any derivatizing reagents?, without derivatization, and using only
366 nm, only UV active compounds are visible and the data provided by authors are
not sufficient to claim that the authors developed a HPTLC method.

Response: We appreciate the reviewers’ comments. The reason why the authors did not
derivatize the HPTLC plate because the species-specific bands of the Mallotus
repandus, Anamirta cocculus and Croton caudatus were simply detected under the
wavelength of 366 nm plus the high quality of HPTLC plate that provides good
separation efficiency, therefore, no further derivatizing reagents is needed in this
study. Moreover, the authors agree that HPTLC method we adapt in this study are not
sufficient to claim that the authors develop a HPTLC method. Therefore, the term
“develop HPTLC” has been change to “HPTLC”.

 

Reviewer #2: 

The present study established a combined Bar-HRM and HPTLC technique to identify the
correct species, Mallotus repandus, from authentic Anamirta cocculus and Croton
caudatus. This method was successfully applied to identify crude drugs and
multiherbal mixed formulae and serve as a quality control tool for preventing
accidental confusion of herbal species sharing the same common name. The experiment
design is logical and fine. However, there are still some issues that merit authors’
attention.

Comments 1: The writing of the “Introduction” is too wordy, especially lines 59-76.
Simply explain that A. cocculus and C. caudatus can't be used as substitutes of M.
repandus in clinical practices.

Response: We appreciate the reviewers’ comments. The authors agree with the reviewer
that the introduction is too wordy. Therefore, the authors rewrite the paragraph to
be more concise. 

Although the A. cocculus and C. caudatus cannot be used as substitutes of M. repandus
in clinical practices but the A. cocculus and C. caudatus also possess their own
medicinal properties. Therefore, the authors would like to use the first paragraph
in the Introduction part to describe their healing properties to mention that they
cannot be used because of the difference in healing properties and the paragraph has
been revised in line 57-74 as following.

Original sentence: 

Line 59-76 of the original manuscript: “However, at least three herbs, namely,
Mallotus repandus (Willd.) Müll. Arg. (Euphorbiaceae), Croton caudatus Gleiseler
(Euphorbiaceae) and Anamirta cocculus (L.) Wight & Arn (Menispermaceae), share
the common name “Kho-Khlan”, and they have different healing properties [4] (Fig 1).
M. repandus is the only official “Kho-Khlan” species and is prescribed as a main
component in the formula. M. repandus has long been used for the relief of muscle
pain [2]. C. caudatus is administered for headaches, visceral pain, and rheumatism
[5]. This species is also reported to treat malaria, fever, numbness, and
constipation [6]. In some parts of Asia, C. caudatus is applied as a poultice to
treat fever and sprains [7]. In addition, the crude extract of C. caudatus seeds can
protect against larvae of mosquitoes [8]. The folk literature indicates that most of
the plants in the genus Croton cause irritation and allergic responses [9]. A.
cocculus is used in the treatment of blood stasis and fever and stimulates the
central nervous system [11]. This species is recorded as a restorative medical herb
in the southern region of Thailand [11]. The plant, however, contains very strong
neurotoxin compounds, such as picrotoxin, picrotin, methyl picrotoxate,
dihydroxypicrotoxinin, picrotoxic acid and a sesquiterpene mixture of picrotoxinin,
that affect the central nervous system (CNS) of vertebrates [12-14]. Seeds of A.
cocculus are used to eliminate unwanted wild fish in aquaculture ponds and to kill
birds [15]. A case study reported that consuming A. cocculus berries caused
extensive brain hemorrhage in cattle [14]. Small amounts of A. cocculus are highly
toxic and fatal if consumed by humans [12]. Although the substances in A. cocculus
are harmful, the herb is still utilized in Thai traditional medicine due to the
belief that a very small dose of toxic substances can be neutralized by other
compounds in the herbal formula [16]”.

Revised sentence: 

Line 57-74 of the revised manuscript: “M. repandus (Euphorbiaceae) shares the common
name “Kho-Khlan” with Croton caudatus Gleiseler (Euphorbiaceae) and Anamirta
cocculus (L.) Wight & Arn (Menispermaceae) (Fig. 1). However, only M. repandus
(Fig. 1A) is an official plant species in NLEM. 

The stem of M. repandus has long been used for the relief of muscle pain in Thai
traditional medicine [2]. C. caudatus is administered for headaches, visceral pain,
rheumatism, fever, and constipation [4-6]. The crude extract of C. caudatus seeds
can protect against mosquito larvae [7]. A. cocculus is used in the treatment of
blood stasis and fever, stimulates the central nervous system [8] and is recorded as
a restorative medical herb in the southern region of Thailand [9]. However, a
previous report showed that C. caudatus causes irritation and allergic responses
[10], while A. cocculus contains very strong neurotoxin compounds that affect the
central nervous system (CNS) of vertebrates, such as picrotoxin, picrotin, methyl
picrotoxate, dihydroxypicrotoxinin, picrotoxic acid and a sesquiterpene mixture of
picrotoxinin [11-13]. Seeds of A. cocculus are also used to eliminate unwanted wild
fish in aquaculture ponds and to kill birds [14]. Consuming A. cocculus berries
causes extensive brain hemorrhage in cattle, while small amounts of A. cocculus are
highly toxic and fatal if consumed by humans [11, 13]. Although the substances in A.
cocculus are harmful, the herb is still utilized in Thai traditional medicine due to
the belief that a very small dose of toxic substances can be neutralized by other
compounds in the herbal formula [15].”

Comments 2: The description of the existing identification methods in the second
paragraph of “Introduction” is logically confused. Please rewrite.

Response: Thank you reviewer for raising this point. The Introduction part was
rewritten to be more logical. The introduction section has been revised in line
80-117 of the revised manuscript as following. 

Original paragraph: 

Line 83-106 of the original manuscript: “Only M. repandus is used for YPSKK
preparation, and it is challenging to differentiate among the three “Kho-Khlan”
species (Fig 1A) when they are in processed forms (Fig 1B-C). Thus, a precise
identification tool is necessary to avoid negative health effects that can occur by
using raw materials from incorrect species. Classical procedures for the
identification of herbs involve organoleptic methods and micro- and macroscopic and
chemical characters [17]. The organoleptic and micro- and macroscopic methods are
basic techniques that require simple equipment and experienced personnel for
taxonomic examination. Thin-layer chromatography (TLC) is used for phytochemical
identification of raw herbal material and is also recommended in the Thai Herbal
Pharmacopoeia [18]. TLC and other tools, such as high-performance TLC (HPTLC) and
high-performance liquid chromatography (HPLC), require standard compounds as
references. Phytochemical composition may be uncertain due to environmental factors.
Although genetic methods based on DNA sequence analysis require specialists and are
cost effective, these methods provide species-level information, and a small number
of samples is needed for the identification process [19]. For a decade, DNA
barcoding has been established and applied for species authentication and
identification. DNA barcoding coupled with high-resolution melting (Bar-HRM)
analysis have gained attention for its fast identification of herbal species such as
Vaccinium myrtillus [20], Mitragyna speciosa [21] and Ardisia gigantifolia [22].
However, Bar-HRM methods have different benefits and limitations; therefore, an
integrative approach is proposed to differentiate substitutions and adulterants of
herbal species [19, 23]. In this study, we aimed to utilize phytochemical profiles
and DNA information to differentiate M. repandus from C. caudatus and A. cocculus.
HPTLC and Bar-HRM approaches were combined to create a simple and fast
identification method for the quality control of “Kho-Khlan” raw material in the
herbal industry.”

Revised paragraph:

Line 80-117 of the revised manuscript: “The stem of M. repandus is used for the YPSKK
formula. Crude drugs of M. repandus are commercially provided in both powdered form
and small pieces of stem, which are challenging for species differentiation (Fig.
1B-C). Although raw materials of herbal medicine can be examined by simple
organoleptic methods and macroscopic and microscopic methods, experienced personnel
for taxonomic examination are required [16]. Thin-layer chromatography (TLC) and
high-performance TLC (HPTLC), which are recommended in the herbal pharmacopoeias of
many countries, including Thailand, are reliable methods for phytochemical
constituent examination; however, the methods require a target compound as a
standard reference [17, 18]. HPTLC, a sophisticated form of TLC, provides good
separation efficiency due to the higher quality of its separation plate. HPTLC
exhibits higher accuracy, reproducibility, and ability to document the results than
TLC [18]. Therefore, this method has been used to determine the phytochemical
profile of herbal species. However, uncertain results may occur due to environmental
factors that affect the chemical composition of herbal species and biological
activities of the substances [19]. In recent years, a molecular approach called the
DNA barcoding technique has gained demand in species identification because it is an
accurate, cost-effective and reliable tool for species identification. The DNA
barcoding method provides species-level information, and small amounts of samples
are needed for the identification process [19].

Currently, DNA barcoding coupled with high-resolution melting (Bar-HRM) analysis has
gained attention for its rapid identification of herbal species such as Vaccinium
myrtillus L. [20], Mitragyna speciosa Korth [21] and Ardisia gigantifolia Stapf
[22]. Bar-HRM, a sequencing-free method, detects signal alterations during the
dissociation of double-stranded DNA generated from the PCR into single-stranded DNA.
Each plant species can be differentiated by their individual melting temperature
(Tm), which is correlated to their nucleotide sequences in the target region [23].
Bar-HRM analysis is a fast, cost-effective and reliable method; moreover, a small
amount of sample is required for species identification. However, Bar-HRM primer
design is challenging when the target sequence has high variation rates across the
target amplicon, and Bar-HRM analysis is limited when low-quality DNA templates are
used [24].

As mentioned above, each identification method has advantages and limitations;
therefore, an integrative approach is proposed to differentiate substitutions or
adulterants of herbal species [19, 25]. Combined phytochemical profiles and DNA
information can be applied to prevent the use of incorrect herbal species and
support the quality of herbal materials to meet international standards [19]. In
this study, we aimed to utilize HPTLC and Bar-HRM analysis to differentiate a pain
relief herb, M. repandus, from C. caudatus and A. cocculus, which share the common
name Kho-Khlan. Combined approaches were used to create a simple and rapid
identification method for the quality control of the Kho-Khlan raw material in the
herbal industry.”

Comments 3: Why did authors select the rbcL region for Bar-HRM analysis for the
differentiation of M. repandus from A. cocculus and C. caudatus? 

Response: The authors selected rbcL gene as a target region for Bar-HRM analysis due
to the sequence alignment result of the DNA barcode regions among M. repandus, A.
cocculus and C. caudatus, we found the rbcL region possess a target site of 102 bp
which contained two conserve regions and one variable region. The 102 bp amplicon
has one conserve region at the 5´- and another at the 3´-end. Nine nucleotide sites
(within the variable region) flanked the two conserve regions and results in the
difference of melting temperature among each 102 bp amplicons of M. repandus, A.
cocculus and C. caudatus. These characters, two conserve regions flank by nucleotide
variable sites and the amplicon site less than 300 bp, suite for Bar-HRM primer
design and only rbcl gene exhibits the characters. 

Therefore, the authors added the reason for the selection of the rbcL region for
Bar-HRM analysis to differentiate M. repandus from A. cocculus and C. caudatus in
line 389-396 and line 400-403 as following. 

Line 389-396 of the revised manuscript: “Since the gene sequences in the rbcL regions
of A. cocculus, C. caudatus and M. repandus possess two conserved sites flanking
nine nucleotide polymorphism sites, this region is suitable for the design of
Bar-HRM primers. The rbcL region was chosen as a targeted amplified region for
Bar-HRM analysis. The nucleotide variation within 102 bp of PCR amplicons amplified
from the three species resulted in different melting temperatures when analyzed by
the Bar-HRM approach. An amplicon of 102 bp is in the range of desired amplicon
lengths for the Bar-HRM analysis (<300 bp) suggested by Osathanunkul et al., 2015
[25].”

Line 400-403 of the revised manuscript: “Moreover, the use of the rbcL region for
species differentiation at the genus level has been revealed [38]. These results
support our conclusion on the reliability of the rbcL region as a potential DNA
barcode marker for discrimination of the nonrelated species that belong to different
genera, A. cocculus, C. caudatus and M. repandus.”

Comments 4: I cannot find the HPTLC bands of C. caudatus (Rf = 0.55) in tracks 6-7 in
figure 3. Besides, the enlarged image on the right in Figure 3 should have a
scale.

Response: Thank you for raising this point. The authors agreed with reviewer that
HPTLC bands of C. caudatus (Rf = 0.55) in tracks 6-7 in figure 3 are missing. In
order to see the missing band, the authors adjusted mobile phase ratio from
toluene:acetone:formic acid, 5:3:0.5 (v/v/v) to 5:4:0.5 (v/v/v). This change
resulted in the Rf alteration of species-specific band from Rf=0.55 to Rf=0.60 (see
the revised Fig. 2). Therefore, the authors changed the original Fig. 3 with the
revised Fig. 3. Scale in revised Fig. 3 has been added in the enlarge image revised
Fig. 3. Please see detail in the "Response to Reviewers" file.

Comments 5: This study showed that compared with A. cocculus, species C. caudatus and
M. repandus were more difficult to distinguish. It is suggested that authors add
some discussion to elaborate on how the two methods, Bar-HRM and HPTLC, complement
each other to distinguish the three species, especially the distinction between C.
caudatus and M. repandus.

Response: Thank you for reviewer’s suggestion. The authors added information in the
discussion section to elaborate on how the two methods, Bar-HRM and HPTLC,
complement each other to distinguish the three species in the discussion part in
line 411-414 as following. 

Line 411-414 of the revised manuscript: “This suggested that Bar-HRM and HPTLC can
complement each other to distinguish C. caudatus and M. repandus when uncertainty in
phytochemical constituents is observed. Bar-HRM analysis using genetic information
can be used to clarify the ambiguous result, as the genetic information is
stable.”

to Reviewers.docx
---

## [Editor Report · Decision Letter 1]

6 May 2022

Combining DNA and HPTLC profiles to differentiate a pain relief herb, Mallotus
repandus, from plants sharing the same common name, “Kho-Khlan”

PONE-D-22-00460R1

Dear Dr. Sukrong,

We’re pleased to inform you that your manuscript has been judged scientifically
suitable for publication and will be formally accepted for publication once it meets
all outstanding technical requirements.

Kind regards,

Branislav T. Šiler, Ph.D.

Academic Editor

PLOS ONE
---

## [Editor Report · Acceptance letter]

2 Jun 2022

PONE-D-22-00460R1 

Combining DNA and HPTLC profiles to differentiate a pain relief
herb, *Mallotus repandus*, from plants sharing the same common
name, “Kho-Khlan” 

Dear Dr. Sukrong:

I'm pleased to inform you that your manuscript has been deemed suitable for
publication in PLOS ONE. Congratulations! Your manuscript is now with our production
department. 

Kind regards, 

on behalf of

Dr. Branislav T. Šiler 

Academic Editor

PLOS ONE